# Targeting IL-21 to tumor-reactive T cells enhances memory T cell responses and anti-PD-1 antibody therapy

Ying Li [1,5✉], Yanni Cong[1,2,5], Mingming Jia [1], Qianqian He[1,2], Haiqing Zhong[1,2], Yun Zhao[3], Hang Li[1], Meining Yan[1], Jia You[1], Jia Liu[1,2], Lieping Chen [4], Haiying Hang[2,3] & Shengdian Wang [1,2✉]

T cell rejuvenation by PD-1/PD-L1 blockade, despite emerging as a highly promising therapy for advanced cancers, is only beneficial for a minority of treated patients. There is evidence that a lack of efficient T cell activation may be responsible for the failure. Here, we demonstrate that IL-21 can be targeted to tumor-reactive T cells by fusion of IL-21 to anti-PD-1 antibody. To our surprise, the fusion protein PD-1Ab21 promotes the generation of memory stem T cells ($T_{SCM}$) with enhanced cell proliferation. PD-1Ab21 treatment show potent antitumor effects in established tumor-bearing mice accompanied with an increased frequency of $T_{SCM}$ and robust expansion of tumor-specific CD8$^+$ T cells with a memory phenotype, and is superior to a combination of PD-1 blockade and IL-21 infusion. Therefore, we have developed a potential strategy to improve the therapeutic effects of immune checkpoint blockade by simultaneously targeting cytokines to tumor-reactive T cells.

[1] CAS Key Laboratory of Infection and Immunity, Institute of Biophysics, Chinese Academy of Sciences, Beijing, China. [2] University of the Chinese Academy of Sciences, Chinese Academy of Sciences, Beijing, China. [3] Key Laboratory of Protein and Peptide Pharmaceuticals, Institute of Biophysics, Chinese Academy of Sciences, Beijing, China. [4] Department of Immunobiology and Yale Cancer Center, Yale University, New Haven, CT, USA. [5] These authors contributed equally: Ying Li, Yanni Cong. ✉email: liy@ibp.ac.cn; sdwang@ibp.ac.cn

PD-1 blockade, which reinvigorates the tumor-reactive CD8[+] T cells by removing the inhibition induced by the interaction of PD-1 and PD-L1, has achieved high success in mediating complete, durable responses in patients with advanced or conventional therapy-resistant cancers; however, the successes are unfortunately limited to a minority of patients[1–3]. A substantial effort is currently underway to identify the factors that determine the therapeutic efficacy of PD-1 blockade and to develop combination therapeutic approaches to increase patient response rates[4–6].

As our understanding of the response or resistance to PD-1 blockade continues to evolve, it is generally recognized that successful antitumor immune responses following PD-1 blockade require reactivation and proliferation of antigen-experienced CD8[+] T cells present in the tumor microenvironment (TME)[4]. In this respect, the $\gamma_c$ family cytokines, especially IL-2, IL-15, and IL-21, play an important role in regulating the magnitude and function of CD8[+] T cell response[7]. IL-2 has been widely used to activate and expand T cells in vitro for adoptive immunotherapy. Compared to IL-2, IL-21 plays a key role in the development and maintenance of memory CD8[+] T cells through the induction of an early differentiation phenotype[8]. IL-21 acts synergistically with IL-7 or IL-15 to promote proliferation and survival of both memory and naïve CD8[+] T cells[9]. More importantly, tumor-reactive T cells generated under the influence of IL-21 show a superior antitumor effect in vivo compared to T cells grown in other γc cytokines[8,10]. Recent studies have shown that programming activated T cells with IL-21 in vitro facilitates the generation of human memory stem-like T cells and improves adoptive immunotherapy[11–13]. In addition, IL-21 has been shown to be an essential component of the CD4[+] T cell help in sustaining the CD8[+] T cell response during chronic, but not acute, lymphocytic choriomeningitis virus (LCMV) infections[14–16]. It was recently reported that combination with IL-21 extended the efficacy of anti-CTLA-4 or anti-PD-1 treatments in preclinical models[17].

For cytokine-based therapies, numerous challenges exist including pharmacokinetic barriers and side effects. Most cytokines, including IL-21, exerts their effects on a broad range of cell types including immune and non-immune cells. In physiological conditions, cytokines limit their actions to specific target cells in a paracrine and autocrine fashion and via short half-life. The systemic administration of cytokines usually has poor therapeutic effects and causes side effects, because the cytokines can also activate counter-regulatory pathways and cause toxicities by acting on different target cells, while activating immune cells to potentiate antitumor immune responses. Therefore, targeting IL-21 to tumor-specific T cells in vivo is potentially invaluable to improving the therapeutic effects of IL-21. PD-1 expression on T cells is induced upon TCR activation. This transient expression decreases in the absence of TCR signaling but is maintained through persistent TCR stimulation, such as in chronic viral infection and in the TME[18,19]. In cancer patients, tumor-specific CD8[+] T cell express a high level of PD-1, and the presence of PD-1 can identify the repertoire of tumor-reactive CD8[+] T lymphocytes[20,21]. Furthermore, only this particular PD-1-positive fraction contains T lymphocytes specific to neoantigens[22] or Melan-A[23]. In comparison to PD-1[-] T cells from the tumor-infiltrating lymphocytes (TILs), T cells from the PD-1[+] fraction exhibited tumor reactivity and resulted in tumor control after being transferred to tumor-bearing mice[24,25]. Furthermore, the presence of PD-1[high] CD8[+] T cell in tumors is predictive for both the response and survival in lung cancer patients treated with PD-1 blockade[26]. Therefore, PD-1 is considered as a marker of activated tumor-reactive T cells[22,23]. We hypothesize that the fusion of IL-21 to anti-PD-1 antibody will concentrate IL-21 to PD-1[+] T cells in vivo, which can greatly increase the effect of IL-21 on tumor-specific T cells while reducing its side effects.

In this study, we developed a fusion protein (PD-1Ab21) of anti-PD-1 antibody and IL-21 that blocked the interaction of PD-1 on T cells with PD-L1 and simultaneously targeted IL-21 to PD-1[+] T cells. PD-1Ab21 stimulated the differentiation of activated T cells back to memory stem T cells (T$_{SCM}$) and had potent antitumor effects in various established tumor models by promoting tumor-specific memory CD8[+] T cell responses.

## Results

**Construction of PD-1Ab21 by fusing anti-PD-1 with IL-21.** In order to develop an anti-PD-1 antibody-based immunocytokine that combines T cell targeting with favorable pharmaceutical properties in vivo, we used the noncovalent homodimeric form ('diabody') of anti-PD-1 single chain antibody fragment (scFv). The diabody moiety, in which the variable heavy and light domains of anti-PD-1 antibody are joined together by a 5-amino acid linker[27], was either sequentially fused with IL-21 and Flag or with Flag alone to produce fusion proteins of PD-1Ab21 or PD-1Ab, respectively (Fig. 1a). Both constructs were cloned into the pTT3 vector and transiently transfected into 293E cells for protein production. After purification on anti-Flag beads, PD-1Ab21 and PD-1Ab showed bands of the expected size on SDS-PAGE (Fig. 1b). Size-exclusion chromatography showed that the noncovalent homodimer was the predominant form of the purified PD-1Ab21 (Fig. 1c).

As expected, PD-1Ab21 bound to PD-1-expressing EG7 lymphoma cells and activated CD8[+] T cells, which completely blocked the binding of PD-L1IgFc. The binding of PD-1Ab21 was blocked by anti-PD-1 antibody (Fig. 1d). Additionally, PD-1Ab21 had the same effect on the proliferation of the pro-B cell line Baf3 as recombinant IL-21 (Fig. 1e). Taken together, these results demonstrate that PD-1Ab21 can bind PD-1 on activated T cells to block the interaction of PD-1 with PD-L1 while maintaining IL-21 bioactivity.

**PD-1Ab21 induces naïve-like T cells differentiated from activated CD8[+] T cells.** Compared to IL-2, IL-21 has been reported to confer a distinct differentiation program on CD8[+] T cells, which is characterized by induction of naïve-like T cells with a CD44[low/intermediate] phenotype[8]. To determine the effect of PD-1Ab21 on the differentiation of antigen-primed CD8[+] T cells, OVA-specific TCR transgenic CD8[+] T (OT-1) cells were activated with OVA peptide and then cultured with different cytokines or proteins. After activation with OVA peptide for two days, OT-1 cells became CD44[high]CD25[+] and expressed high levels of PD-1 (Supplementary Fig. 1a). As expected, after differentiation with cytokines, the IL-2-cultured T cells displayed the CD44[high] phenotype, while T cells cultured in IL-21 were less differentiated with one third of cells displaying the CD44[low]CD62L[high] naïve phenotype. Interestingly, more than half of PD-1Ab21-cultured T cells have the naïve phenotype. PD-1Ab had no effect on T cell differentiation (Fig. 2a). Similar results were also obtained from the polyclonal CD8[+] T cells activated with anti-CD3 and anti-CD28 antibodies (Supplementary Fig. 1b).

The CD44[low]CD62L[high] cell population cultured in PD-1Ab21 could potentially be the remaining naïve cells after antigen encounter, or they could have potentially been converted back to the naïve phenotype from activated CD44[high] T cells. To distinguish between these two possibilities, the CD44[high]CD62L[high] population was isolated by sorting post activation for 2 days (Supplementary Fig. 1c), and then cultured with PD-1Ab21 or medium alone. During the 3-day culture, the control cells cultured in medium alone maintained CD44[high]CD62L[high] phenotype, while the cells cultured with PD-1Ab21 gradually became CD44[low]CD62L[high] phenotype. The number of CD44[low]CD62L[high] cells gradually increased and

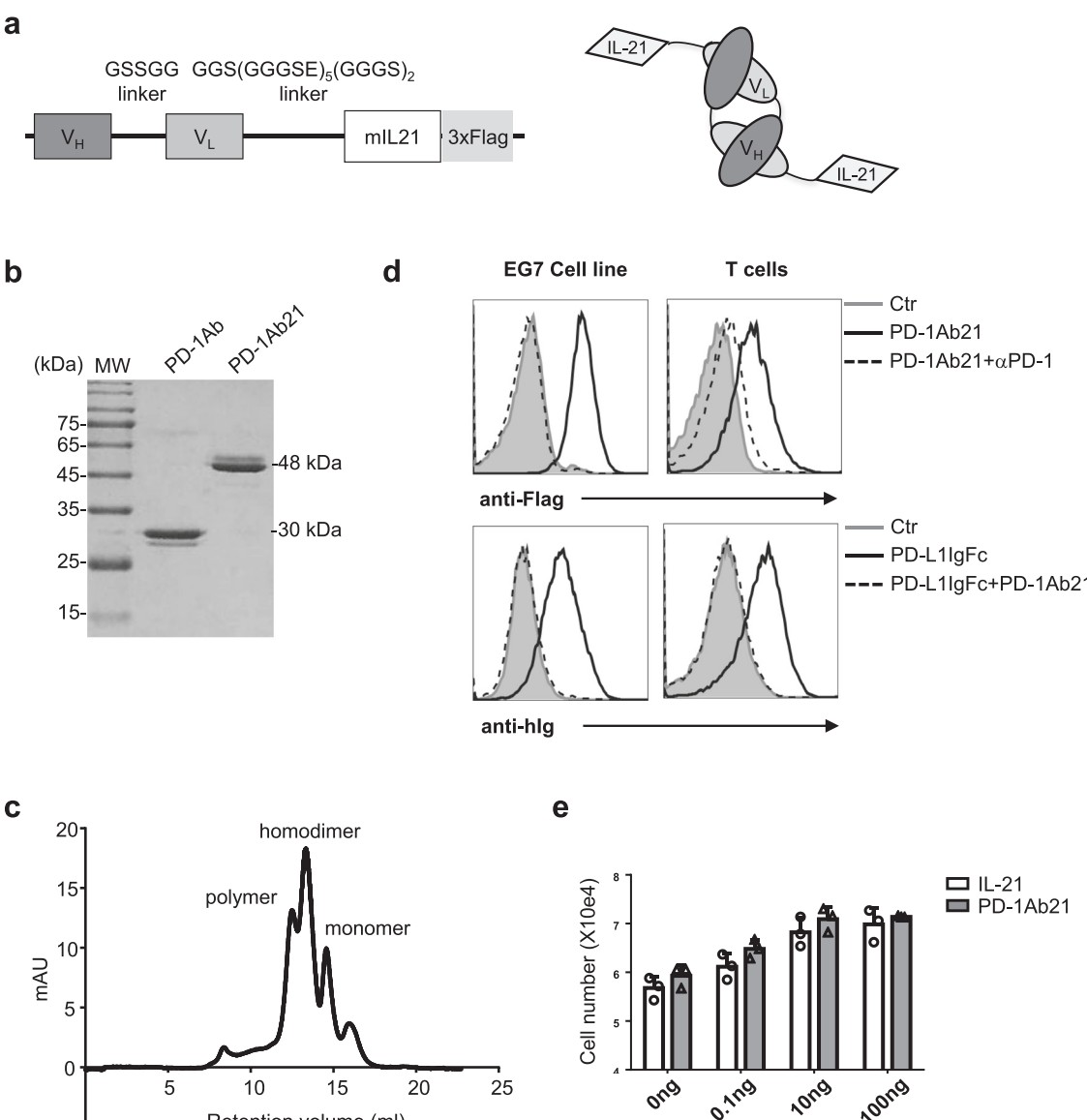

**Fig. 1 Cloning, expression, and characterization of PD-1Ab and PD-1Ab21. a** Schematic of the cloning strategy (left) and domain assembly (right) of PD-1Ab21. **b** SDS-PAGE analysis of the purified proteins PD-1Ab (30 kDa) and PD-1Ab21 (48 kDa). MW, molecular weight. **c** Analytical size-exclusion chromatography of PD-1Ab21. Expected elution volumes were as follows: monomer, 15 ml; dimer, 13 ml; polymer, 12 ml. Protein elution was monitored by measuring absorbance at 280 nm. **d** Binding of PD-1Ab21 to PD-1$^+$ cells. EG7 (left) or activated OT-1 cells (right) were stained without (as control, filled grey histogram) or with PD-1Ab21 in the presence (dashed line histogram) or absence (black line histogram) of anti-PD-1 antibody, and then followed by staining with anti-flag antibody (top). Or the cells were stained without (as control, filled grey histogram) or with PD-L1IgFc in the presence (dashed line histogram) or absence (black line histogram) of PD-1Ab21, and then followed by staining with anti-hIg antibody (bottom). **e** Detection of IL-21 bioactivities. Baf3 cells were cultured with medium only (as control), IL-21 or PD-1Ab21 for 3 d and counted using a cell counter ($n = 3$ replicates /group). Results are presented as mean ± SEM and are representative of more than three independent experiments.

reached its peak after two days in the culture with PD-1Ab21 (Fig. 2b). Next, the sorted CD44$^{high}$CD62L$^{high}$ cells were labeled with CFSE and cultured with either PD-1Ab21 or IL-2. In the culture with PD-1Ab21, the CD44$^{low}$CD62L$^{high}$ population underwent more division than the CD44$^{high}$CD62L$^{high}$ population. As expected, IL-2 differentiated the activated T cells to CD44$^{high}$CD62L$^{low}$ effector or effector memory T cells (T$_E$/T$_{EM}$) by driving robust proliferation across all cell populations. (Fig. 2c). When the activated CD44$^{high}$CD62L$^{high}$ population was further sorted into CFSE$^{high}$ (less division) and CFSE$^{low}$ (more division) before being cultured with PD-1Ab21, both populations resulted in similar division and differentiation patterns (Supplementary Fig. 1d). These results demonstrate that PD-1Ab21 can drive the

differentiation of activated CD44$^{high}$CD62L$^{high}$ CD8$^+$ T cells back to the naïve-like CD44$^{low}$CD62L$^{high}$ phenotype through cell proliferation. However, PD-1Ab21 could not reverse the differentiation of activated CD44$^{high}$CD62L$^{low}$ CD8$^+$ T cells (Supplementary Fig. 1e).

**Targeting PD-1Ab21 to activated T cells induces differentiation of T$_{SCM}$-like cells.** A population of CD44$^{low}$CD62L$^{high}$ CD8$^+$ T cells expressing Sca1-1, CD122, and Bcl-2 was previously designated as T$_{SCM}$ in a mouse model of graft-versus-host disease[28] and in vitro activation of CD8$^+$ T cells with induction of Wnt or Notch signaling[29]. Compared with naïve T cells (T$_N$), PD-1Ab21-

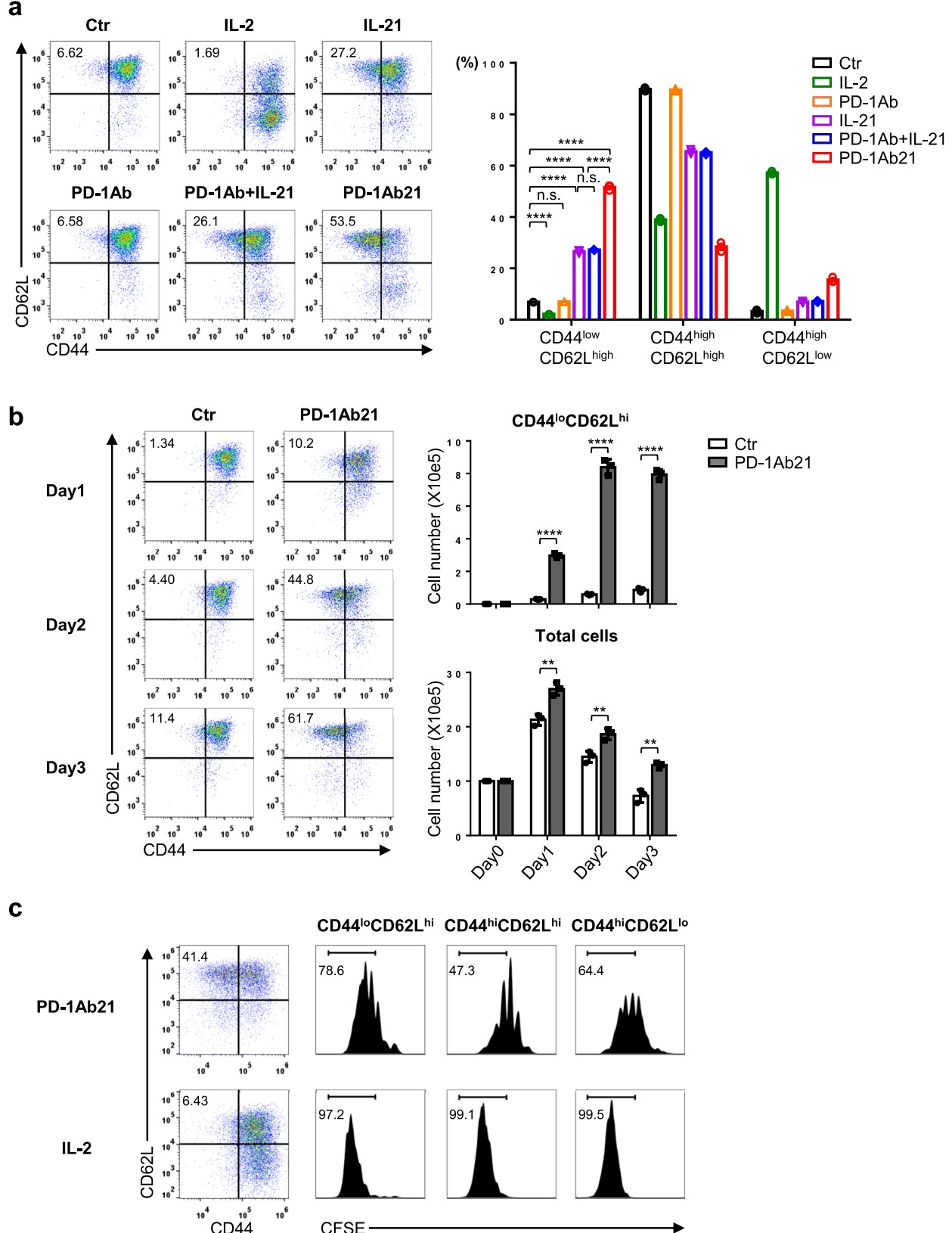

generated CD44$^{low}$CD62L$^{high}$ T cells expressed CD25 and a higher level of Sca1-1 and CD122, which was more similar to IL-15-generated central memory T cells (T$_{CM}$) than IL-2-differentated T$_E$/T$_{EM}$. Bcl-2 was rarely detected across cell populations in our experiments (Supplementary Fig. 2a). To further characterize the CD44$^{low}$CD62L$^{high}$ T cells differentiated with PD-1Ab21, we performed a transcriptome analysis of isolated T$_N$, activated and PD-1Ab-cultured CD44$^{high}$CD62L$^{high}$ T cells, IL-2-generated T$_E$/T$_{EM}$, IL-15-generated T$_{CM}$ as well as PD-1Ab21 and IL-21-generated CD44$^{low}$CD62L$^{high}$ T cells (Supplementary Fig. 2b, c). A total of 9823 differentially expressed genes were identified (Benjamini–Hochberg adjusted $P$-value < 0.01, fold change > 2).

**Fig. 2 PD-1Ab21 promotes the differentiation of naïve-like T cells from activated CD8$^+$ T cells. a, b** Flow cytometry analysis of the differentiation of activated OT-1 cells. Lymph node cells from OT-1 mice were primed with OVA$_{257-264}$ for 2 d. **a** Three days after differentiation with the indicated proteins, T cell differentiation was analyzed by the expression of CD62L and CD44 on gated live CD8$^+$ T cells (left). Column graphs show the relative frequencies of different cell populations (right). **b** The activated CD44$^{high}$CD62L$^{high}$ CD8$^+$ T cells were sorted and differentiated with PD-1Ab21 or medium alone for 3 days. Cells were counted and analyzed by flow cytometry (left). Column graphs show cell numbers of different cell populations (right). **c** Proliferation of differentiated cells. Activated CD44$^{high}$CD62L$^{high}$ CD8$^+$ T cells were sorted and labeled with CFSE. Three days after differentiation with PD-1Ab21, cell proliferation was measured by CFSE dilution. Numbers show the percentage of CD44$^{low}$CD62L$^{high}$ cells in the dot-plot quadrants (left) or the percentages of cells with more than two divisions (right). All data are representative of at least three independently performed experiments. Results are presented as mean ± SEM, and were compared by unpaired two-tailed t-test. **a** ****p < 0.0001, **b** ****p < 0.0001, **p = 0.0033 (d1), **p = 0.0086 (d2), **p = 0.0015 (d3), n.s. not significant.

Principal component analysis revealed that T cells generated by PD-1Ab, PD-1Ab21, or IL-21 clustered together with T$_{CM}$ and T$_N$ cells, while activated T cells were more closely related to T$_E$/T$_{EM}$ (Fig. 3a). Unsupervised hierarchical clustering of T$_N$, T$_{CM}$, and T cells generated by PD-1Ab, PD-1Ab21, or IL-21 showed that both CD44$^{low}$CD62L$^{high}$ T cells generated by PD-1Ab21 or IL-21 were partitioned in a branch of the dendrogram, while PD-1Ab-generated CD44$^{high}$CD62L$^{high}$ T cells were partitioned in the branch of T$_{CM}$ (Fig. 3b). Compared to T$_N$, T$_{CM}$ and PD-1Ab-generated CD44$^{high}$CD62L$^{high}$ T cells, CD44$^{low}$CD62L$^{high}$ T cells generated with PD-1Ab21 or IL-21 were characterized by the highest expression Ly6a (Scal-1), Fas (CD95), Bcl-6[30], STAT3[31,32], and Tcf7(TCF1)[33], which are known to be highly expressed in T$_{SCM}$ or promote memory CD8$^+$ T cell formation (Supplementary Fig. 2d). CD95 is a typical marker of human T$_{SCM}$ that has not been reported in mouse T$_{SCM}$[34,35]. The expression of TCF1 in CD8$^+$ TILs is essential for the generation of effective antitumor immunity in response to diverse immunotherapies, including PD-1 blockade[36,37]. Flow cytometry also showed similar expression patterns of CD95 and TCF1 in these cell populations (Fig. 3c). Thus, the gene profiling analysis provides corroborating evidence that PD-1Ab21 induces conversion of activated CD8$^+$ T cells back to a memory subset that is distinct from T$_{CM}$ cells, but similar to T$_{SCM}$.

To define the functions of the differentiated cells, we assessed their ability to produce cytokines. Both CD44$^{low}$CD62L$^{high}$ cells differentiated with IL-21 and PD-1Ab21secreted high levels of IL-2 and low levels of IFN-γ after being exposed to PMA/ionomycin for 4 h; whereas, IL-2-differentiated effector T cells embodied the opposite pattern (Fig. 3d). The same results were obtained when PD-1Ab21-cultured cells were compared to IL-2-differentiated cells via intracellular staining (Supplementary Fig. 2e). Hence, similar to IL-21, PD-1Ab21 inhibits the effector function of activated T cells, but endows them with the ability to produce IL-2. IL-2 production is a specific property of long-lived antigen-specific CD8$^+$ T cells throughout the differentiation course of CD8$^+$ T cell memory[38].

The aforementioned results showed that PD-1Ab21 was superior to IL-21 for differentiation of T$_{SCM}$-like cells (Fig. 2a). Next, we compared in detail the dose-dependent effects of PD-1Ab21 and IL-21, which were simultaneously prepared, on the differentiation of T$_{SCM}$-like cells. The superior activity of PD-1Ab21 to induce differentiation of T$_{SCM}$-like cells may result from various mechanisms, including the dimeric nature of PD-1Ab21, which should be more active than monomeric IL-21; or enrichment of IL-21 on the T cell surface by binding of PD-1Ab21 to PD-1. Although IL-21IgFc was indeed more effective at stimulating differentiation of T$_{SCM}$-like cells than was recombinant monomer IL-21(Supplementary Fig. 3a), PD-1Ab21 was still much more potent than IL-21IgFc (Fig. 4a). There was minimal difference between the ability of PD-1Ab21 and IL-21IgFc to induce differentiation of PD-1$^{-/-}$ T cells activated by anti-CD3 and anti-CD28, but PD-1Ab21 was markedly superior to IL-21IgFc for differentiation of T$_{SCM}$-like cells from wild-type T cells (Fig. 4b). This data suggest that PD-1Ab21 promotes

differentiation of T$_{SCM}$-like cells from activated T cells through two effects: dimerization and targeting. The differentiation of T$_{SCM}$-like cells induced by IL-21 or PD-1Ab21 can be inhibited by IL-21RIgFc fusion protein (Supplementary Fig. 3b), suggesting that the differentiation of T$_{SCM}$-like cells induced by PD-1Ab21 was mediated by IL-21 receptor.

**PD-1Ab21 enhances the antitumor effects of anti-PD-1 antibody.** To evaluate the antitumor effect of PD-1Ab21, we first used the CT26 and MC38 tumor model, which responded to anti-PD-1 blockade. Tumor-bearing mice were treated with intraperitoneal (i.p.) injections of anti-PD-1 antibody, an equimolar mixture of PD-1Ab and IL-21, or PD-1Ab21 alone. Compared to the non-treated control group, some antitumor effects were observed in the group treated with anti-PD-1 in CT26 tumor model, and the groups treated with anti-PD-1 or PD-1Ab combined with IL-21 in the MC38 tumor model. The PD-1Ab21 treatment exhibited more potent antitumoral effects through repeated experiments in both tumor models (Fig. 5a and Supplementary Fig. 4a). Similar results were also achieved through hydrodynamic injections of the corresponding plasmids (Supplementary Fig. 5a).

We previously showed that anti-Her2/neu Ab treatment activated the antitumor T cell response[39]. We hypothesized that PD-1Ab21 would enhance the therapeutic effect of anti-Her2/neu antibody by targeting tumor-specific T cells activated by anti-Her2/neu therapy. In the case of large TUBO tumors, which is a spontaneous carcinoma derived from neu-transgenic mice, anti-Her2/neu treatment alone showed limited antitumor effect. When anti-Her2/neu was combined with systemic injections of PD-1Ab and IL-21, the therapeutic effects of the antibody slightly increased. However, the combination of the antibody with PD-1Ab21 had a significant inhibitory effect on the growth of all tumors. (Fig. 5b and Supplementary Fig. 4b). The same results were also obtained by combining anti-Her2/neu treatment with hydrodynamic injections of the corresponding plasmids (Supplementary Fig. 5b).

Given the potential to both generate new antigen-specific T cell responses and amplify existing responses against tumor cells, cancer vaccines may be an effective combinational partner with PD-1 blockade[40,41]. To evaluate the synergistic effects of vaccination and PD-1Ab21, B16-OVA-bearing mice, which were adoptively transferred with OT-1 cells 1 day before tumor inoculation, were first vaccinated by intradermal (i.d.) injection of poly I:C and OVA peptide and then treated with PD-1Ab21, anti-PD-1 antibody, or a combination of PD-1Ab and IL-21. Vaccination dramatically delayed tumor growth. Combining the vaccine with PD-1Ab plus IL-21 or anti-PD-1 antibody did not significantly improve its therapeutic effect; however, the PD-1Ab21 treatment markedly enhanced the therapeutic effect of the vaccination (Fig. 5c and Supplementary Fig. 4c). Similar results were obtained by combining a vaccination of CFA and OVA peptide (Supplementary Fig. 5c). Collectively, these data

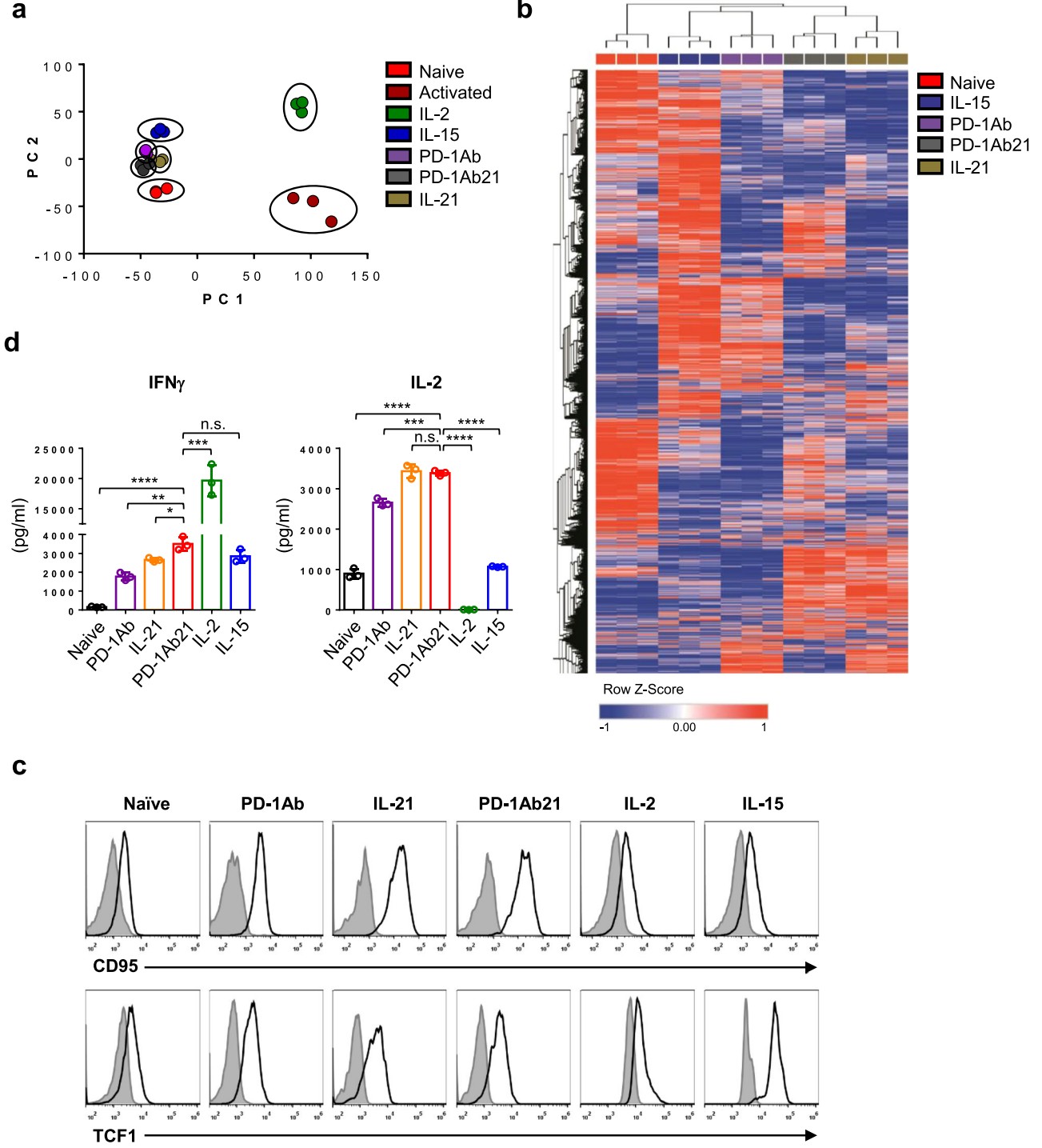

**Fig. 3 Analysis of transcriptome profile and functions of PD-1Ab21-induced naïve-like T cells. a**, **b** RNA-seq was performed in sorted $T_N$, activated CD44$^{high}$CD62L$^{high}$, PD-1Ab21-generated CD44$^{low}$CD62L$^{high}$ and IL-15 or IL-2-differentiated cells. **a** Principal component analysis (PCA) of T cell subsets based on the global transcriptome (~10,000 genes), ($n = 3$ samples for each cell population). **b** Heatmap showing the z-transformed expression values of library size-normalized read counts for each differentially expressed gene, colored from blue to red. RNA-Seq data from each independent sample are shown in the columns. Comparisons of gene expression between each pair of CD8$^+$ T cell subsets were done by DESeq2 analysis (Benjamini–Hochberg adjusted *P*-value < 0.01, fold change > 2). **c** Flow cytometry analysis for expression of Fas and TCF1 in $T_N$ and cells differentiated with different proteins. The isotype (filled histogram) and anti-CD95 (top) or anti-TCF1 (bottom) (open histogram) antibody staining are shown. **d** Cytokine production of T cells. T cell subsets were sorted as described in **a**, and stimulated with PMA/ionomycin for 4 h. IFN-γ and IL-2 in the supernatants were detected by ELISA. The data are presented as mean ± SEM and are representative of at least three independently performed experiments. Results were compared by unpaired two-tailed *t*-test. **d** *$p$ = 0.0165, **$p$ = 0.0019, ***$p$ = 0.0004 (left and right panel), ****$p$ < 0.0001, n.s. not significant.

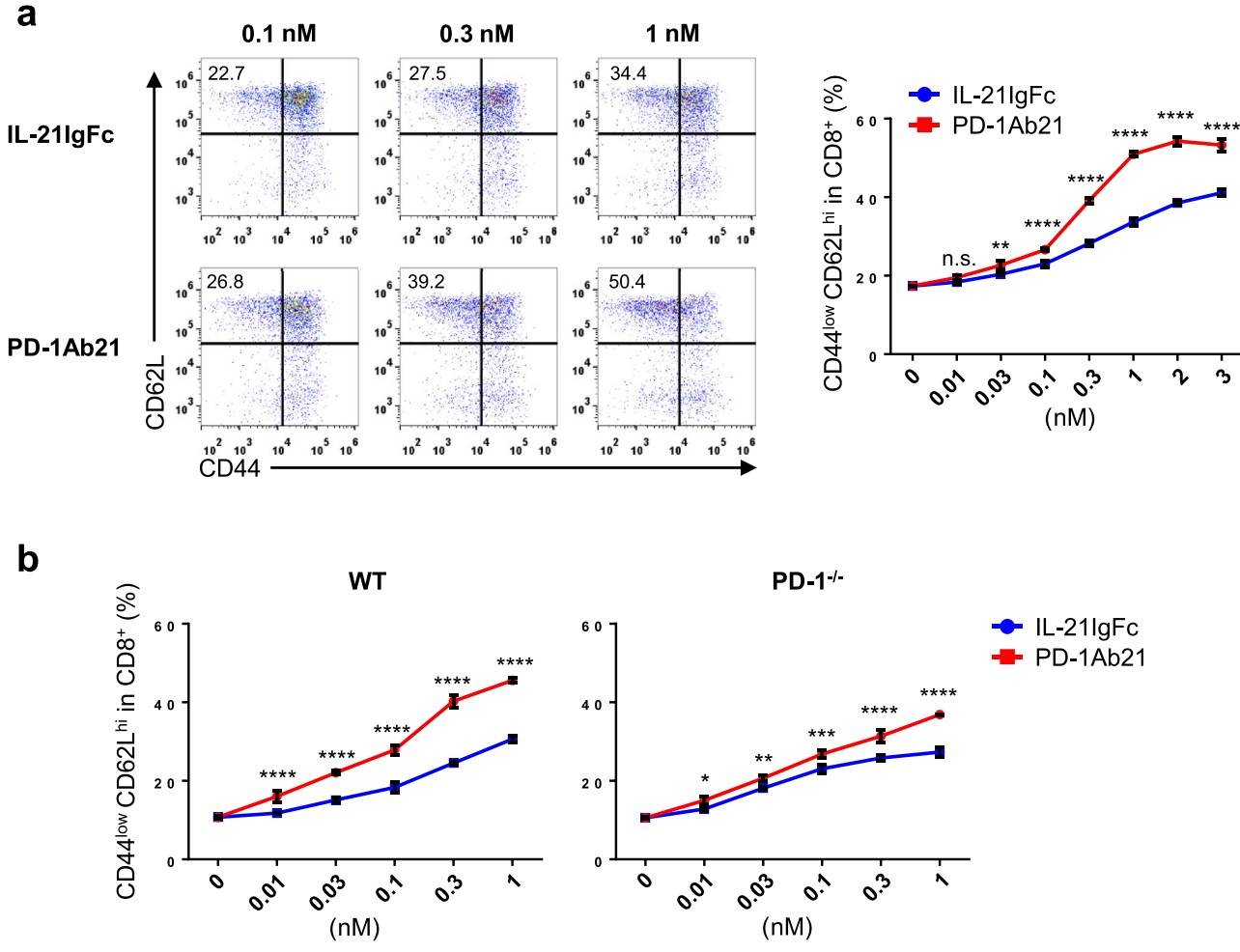

**Fig. 4 Targeting effects of PD-1Ab21 on differentiation of T$_{SCM}$-like cells by binding to PD-1$^+$ T cells. a, b** Naïve OT-1 cells were activated by OVA$_{257-264}$ peptide for 2 d and then differentiated with different concentrations of cytokines or proteins for 3 d. T cell differentiation was analyzed by flow cytometry. **a** Representative dot-plots of T$_{SCM}$ differentiation with different concentrations of IL-21IgFc and PD-1Ab21 (left). Percentage of CD44$^{low}$CD62L$^{high}$ CD8$^+$ T cells was counted (right). **b** Naïve T cells isolated from LNs of WT or PD-1$^{-/-}$ mice were primed with anti-CD3 and anti-CD28 for 2 d, and then differentiated with the indicated concentrations of IL-21IgFc or PD-1Ab21. Percentage of CD44$^{low}$CD62L$^{high}$ CD8$^+$ T cells was counted. All data are presented as mean ± SD and representative of at least three independently performed experiments. Results were compared using two-way ANOVA followed by Tukey's multiple comparison test. **a** **$p = 0.0086$. ****$p < 0.0001$. **b** *$p = 0.0257$, **$p = 0.0082$, ***$p = 0.0001$, ****$p < 0.0001$, n.s. not significant.

suggest that the fusion protein PD-1Ab21 has superior antitumor activities to the combination of PD-1 blockade and IL-21.

**PD-1Ab21 treatment expands tumor-specific memory CD8$^+$ T cells.** To detect the targeting of PD-1Ab21 to tumor-specific T cells in vivo, we assessed PD-1-receptor occupancy by the injected PD-1Ab21 on OT-1 cells in the B16-OVA tumor model. OT-1 cells expressed high levels of PD-1 in vaccinated mice. At 30 min after the injection of PD-1Ab21 or anti-PD-1 antibody, PD-1 could not be detected on OT-1 cells from the peripheral blood of the treated mice via flow cytometry. This suggests that OT-1 cells were bound by injected PD-1Ab21 or anti-PD-1 antibody in vivo. OT-1 cells in tumors were bound a little later. At 24 h, PD-1 was partly detected on OT-1 cells from blood of PD-1Ab21-treated mice, but was not detected on OT-1 cells from anti-PD-1-treated mice (Fig. 6a and Supplementary Fig. 6a). To directly detect the targeting of PD-1Ab21 to T cells in vivo, an immunofluorescence analysis was performed on tumor sections. CD8$^+$ T cells were observed to be co-stained with anti-flag antibody in tumors from mice treated with PD-1Ab21 (which has

a flag tag), but not in tumors from mice treated with recombinant IL-21Flag (Supplementary Fig. 6b). These data demonstrate that PD-1Ab21 rapidly bound to activated tumor-specific T cells after injection.

Next, we systemically analyzed leukocytes in treated tumor-bearing mice. An increase of CD8$^+$ T$_{SCM}$ was observed in spleens and tumor draining lymph nodes (DLNs) of mice treated with PD-1Ab21 (Fig. 6b and Supplementary Fig. 7a, b). Additionally, a decreased frequency of macrophages was observed in tumors treated with PD-1Ab21 compared to tumors treated with anti-PD-1 antibody and IL-21 (Supplementary Fig. 7c). No significant differences were observed in other cell populations. These data confirm that PD-1Ab21 targeted to tumor-reactive T cells to promote differentiation of T$_{SCM}$ in vivo.

For tumor-specific T cell responses, we analyzed OT-1 cells in the B16-OVA tumor model. Compared with vaccine alone, anti-PD-1 did not further increase the OT-1 frequency, suggesting that activated OT-1 cells had not been suppressed by PD-1 signaling. The frequencies of the OT-1 cells were slightly higher in mice treated with vaccination plus PD-1Ab and IL-21 than in mice treated with vaccination alone. In contrast, the combination with

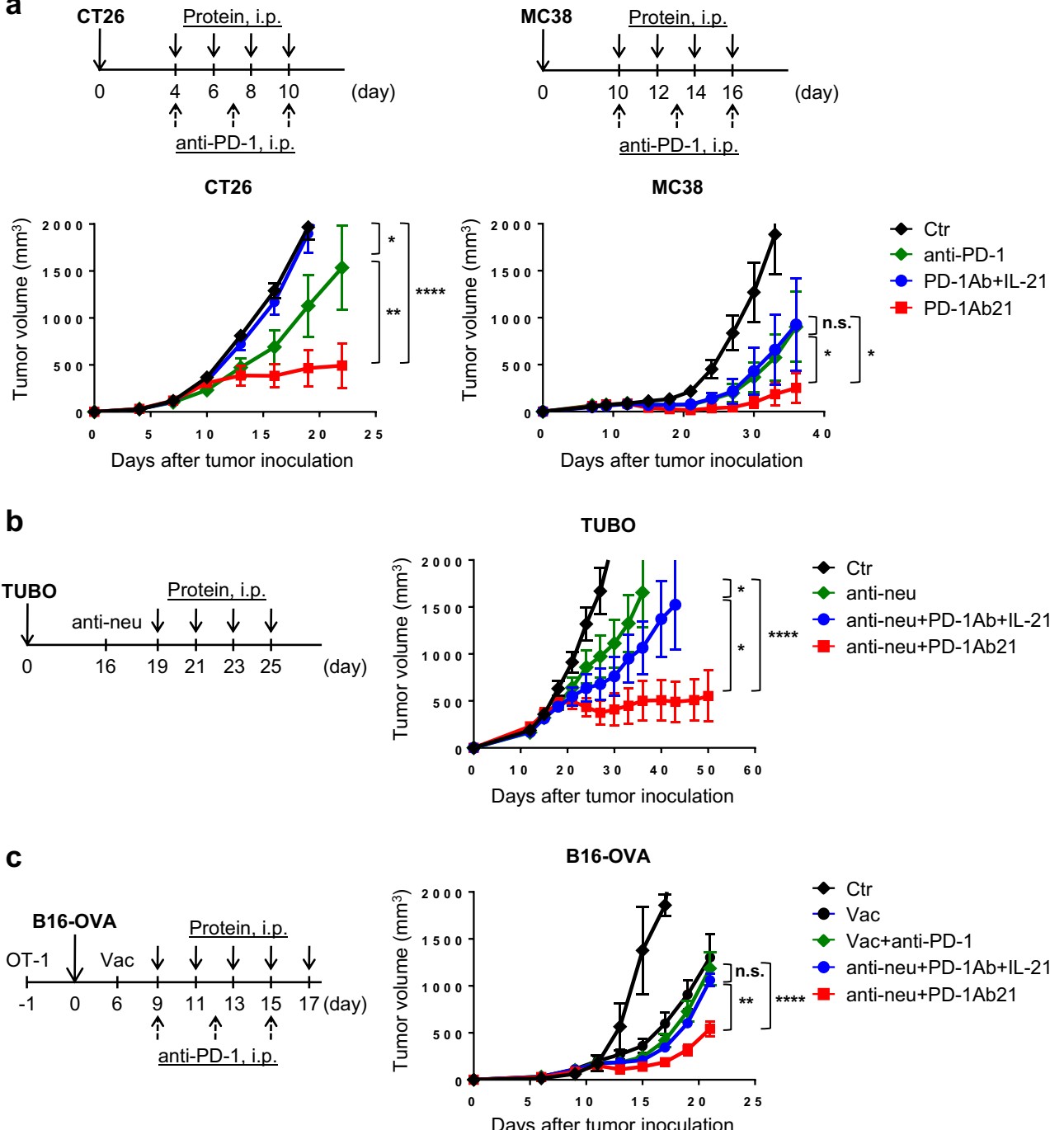

**Fig. 5 Superior antitumor effect of PD-1Ab21 treatment. a** Balb/c mice transplanted s.c. with CT26 cells (left) or C57BL/6 mice transplanted s.c. with MC38 cells (right) were treated with anti-PD-1, PD-1Ab + IL-21, or PD-1Ab21 ($n = 5$ mice /group). **b** Balb/c mice ($n = 5$ mice /group) transplanted s.c. with TUBO cells were treated with anti-Her2/neu antibody alone or in combination with PD-1Ab + IL-21 or PD-1Ab21. **c** C57BL/6 mice ($n = 5$ mice /group) transferred with $1 \times 10^6$ naïve $CD90.1^+$ OT-1 cells 1 d before inoculation of B16-OVA cells were immunized with poly I:C and $OVA_{257-264}$ peptide, followed by treatment with anti-PD-1, PD-1Ab + IL-21, or PD-1Ab21. Tumor length (a) and width (b) were measured and tumor volume was calculated as ($ab^2/2$). Tumor growth is shown as mean tumor size ±SEM over time. Results were compared using two-way ANOVA followed by Tukey's multiple comparison test. **a** CT26: $*p = 0.0167$, $**p = 0.0024$, $****p < 0.0001$; MC38: $*p = 0.0445$ (anti-PD-1 vs. PD-1Ab21), $*p = 0.0242$ (PD-1Ab+IL-21 vs. PD-1Ab21). **b** $*p = 0.0223$ (anti-neu vs. anti-neu+PD-1Ab+IL-21), $*p = 0.0335$ (anti-neu+PD-1Ab+IL-21 vs. anti-neu+ PD-1Ab21), $****p < 0.0001$. **c** $**p = 0.0015$, $****p < 0.0001$, n.s. not significant. One representative experiment out of three independent experiments is shown.

PD-1Ab21 further stimulated the robust expansion of OT-1 cells, leading to a systemic high frequency of OT-1 cells, especially in tumors. The frequency of OT-1 cells in TILs of PD-1Ab21-treated mice was about two times higher than that of anti-PD-1-treated mice, which accounted for more than 70% of $CD8^+$ TILs (Fig. 6c

and Supplementary Fig. 8a, b). However, the non-OT-1 $CD8^+$ T cells did not increase in tumors of PD-1Ab21-treated mice. (Supplementary Fig. 8c). The frequency of the OT-1 cells did not increase in DLNs, but dramatically increased in tumors of mice treated with vaccination or vaccination plus PD-1 blockade

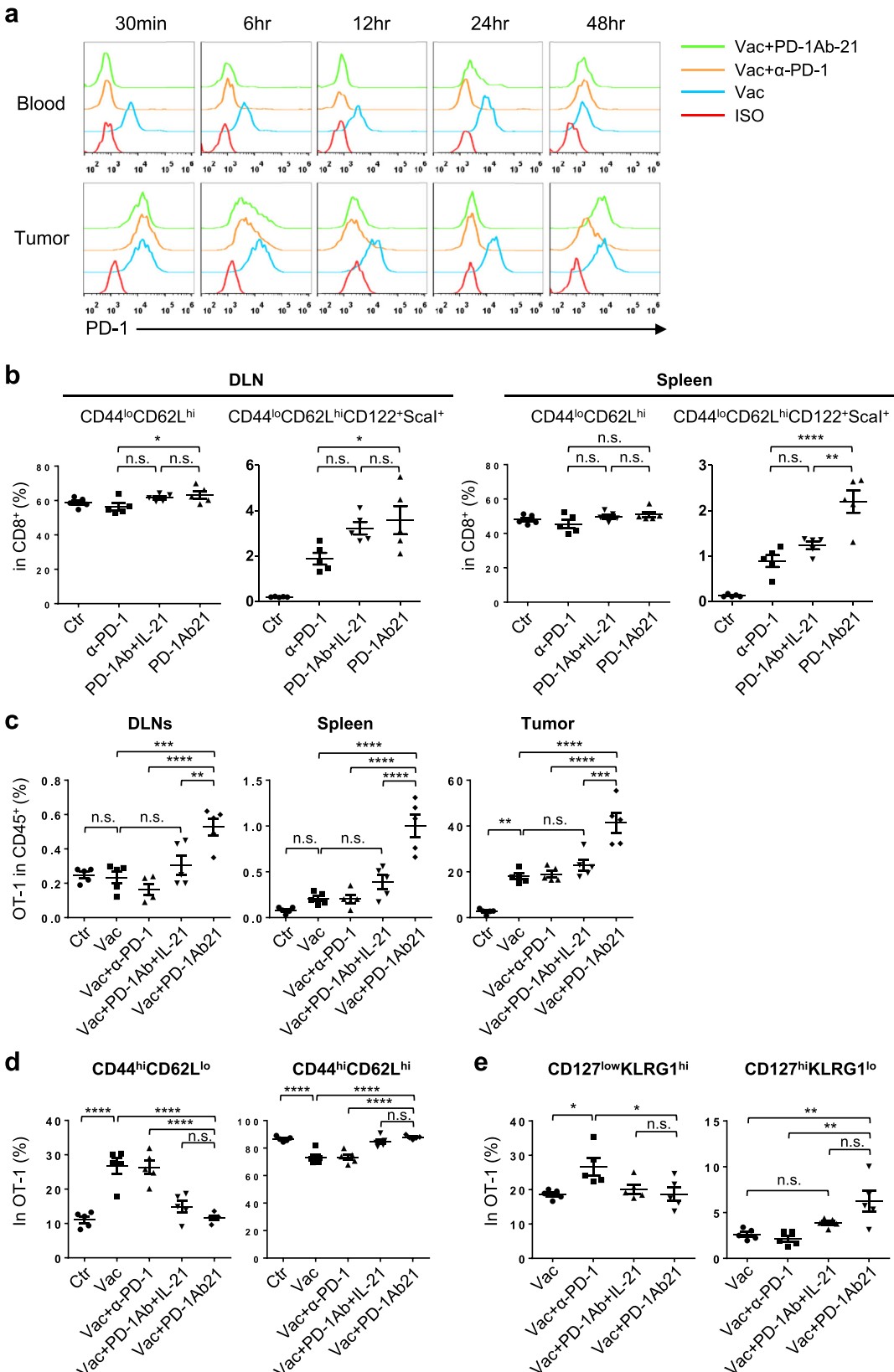

or/and IL-21 compared to that of mice without vaccination. However, the frequency of OT-1 cells was still significantly higher in DLNs of PD-1Ab21-treated mice (Fig. 6c and Supplementary Fig. 8b). This may be due to the fact that activated antigen-specific T cells egressed from DLNs in mice treated with vaccination or vaccination plus anti-PD-1 at this stage after

vaccination, whereas PD-1Ab21 treatment generated resident self-renewing memory T cells in DLNs. This is further supported by the phenotypes of OT-1 cells in DLNs. The frequencies of CD44$^{high}$CD62L$^{low}$ cells in the OT-1 cells were much higher in mice treated with vaccination alone or combination with anti-PD-1 antibody than in mice treated with PD-1Ab21. The OT-1

**Fig. 6 The mechanisms underlying the antitumor effect of PD-1Ab21 treatment. a** PD-1-receptor occupancy by anti-PD-1 antibody and PD-1Ab21 in blood and tumor. B16-OVA tumor-bearing mice with transferred OT-1 cells were immunized with poly I:C and OVA$_{257-264}$ peptide on day 6, followed by i.p. injections of anti-PD-1 or PD-1Ab21 on day 9 after tumor inoculation. Blood and tumors were harvested and the binding of a fluorescent-conjugated anti-PD-1 antibody to OT-1 cells was analyzed by flow cytometry at the indicated time post injections of antibody or PD-1Ab21. One representative experiment out of three independent experiments is shown. **b** Percentage of CD8$^+$ T cells with naïve phenotype and T$_{SCM}$ in DLNs and spleens of CT26 tumor-bearing mice in different treatment groups as described in Fig. 5a ($n = 5$ mice /group). Mice were sacrificed 1 d after last injections of antibody or proteins, and lymphocytes were prepared from DLNs and spleens for flow cytometry analysis. **c–e** B16-OVA-bearing mice with transferred OT-1 cells were immunized with OVA$_{257-264}$ peptide/ poly I:C, followed by treatment with anti-PD-1, PD-1Ab21 or combination of IL-21 and PD-1Ab, and then sacrificed 1 d after last injections of antibody or protein ($n = 5$ mice /group). DLNs, spleens, and tumors were harvested for flow cytometry analysis. **c** Frequencies of OT-1 cells in DLN, spleen and tumor. **d** Phenotypes of OT-1 cells in DLNs. **e** Phenotypes of OT-1 cells in TILs. Data are presented as mean ± SEM and are representative of at least three independent experiments. Results were compared using one-way ANOVA followed by Tukey's multiple comparison test. **b** *$p = 0.0208$ (DLN CD44$^{lo}$CD62L$^{hi}$), *$p = 0.0431$ (DLN CD44$^{lo}$CD62L$^{hi}$CD122$^+$Sca1$^+$), **$p = 0.0013$. **c** DLN: **$p = 0.007$, ***$p = 0.0004$; Tumor: ***$p = 0.0002$. **e** *$p = 0.0214$ (Vac vs. Vac+α-PD-1), *$p = 0.0234$ (Vac+α-PD-1 vs. Vac+PD-1Ab21), **$p = 0.0035$ (Vac vs. Vac+PD-1Ab21), **$p = 0.0012$ (Vac+α-PD-1 vs. Vac+PD-1Ab21). ****$p < 0.0001$, n.s. not significant.

cells were predominantly of CD44$^{high}$CD62L$^{high}$ phenotype in DLNs of PD-1Ab21-treated mice (Fig. 6d). Although the frequency of CD44$^{high}$CD62L$^{high}$ cells in the OT-1 cells in mice treated with IL-21 and PD-1Ab was similar to that in mice treated with PD-1Ab21 (Fig. 6d), OT-1 cells in DLNs of mice treated with IL-21 and PD-1Ab was significantly less than that in mice treated with PD-1Ab21 (Fig. 6c and Supplementary Fig. 8b). These results collectively demonstrate that PD-1Ab21 promoted the generation of T$_{SCM}$ and T$_{CM}$ in DLNs and boosted continuous proliferation and differentiation of tumor-specific CD8$^+$ T cells under the stimulation of peripheral antigens.

As expected, tumor-infiltrating CD8$^+$ T cells, including non-OT-1 CD8$^+$ T cells, predominantly had the CD62L$^{low}$ phenotype. Compared to the anti-PD-1 antibody treatment, the PD-1Ab21 treatment increased the percentage of CD127$^{high}$KLRG1$^{low}$ memory precursor effector cells (MPEC) and decreased the percentage of CD127$^{low}$KLRG1$^{high}$ effector cells (Fig. 6e and Supplementary Fig. 8d). No significant differences were observed for the frequency of NK cells and CD4$^+$ T cells in TILs of PD-1Ab21-treated mice compared with that of mice treated with vaccination or vaccination plus anti-PD-1 (Supplementary Fig. 8c). These results demonstrate that anti-PD-1 only reinvigorates tumor-specific T cells, whereas PD-1Ab21 simultaneously targets IL-21 to tumor-reactive T cells, thus improving the differentiation of memory T cells to enhance T cell responses against tumors.

## Discussion

Antibody-based immunocytokines have been widely used for targeting cytokines to tumors[42,43]. Here, we engineered an anti-PD-1 antibody-based immunocytokine, denoted as PD-1Ab21, by fusing IL-21 to anti-PD-1 diabody to illustrate that cytokines can be targeted to tumor-reactive T cells in vivo to promote cancer immunotherapy.

Tumor-reactive T cells are key mediators of tumor destruction. The differentiation status of T cells is a key characteristic that determines their immune functionality and longevity[44]. T$_{SCM}$ cells mediate highly effective tumor regression in vivo due to their robust proliferative potential after exposure to antigen, long-term survival capacity, and ability to give rise to all memory and effector T cell subsets[29,34,45]. In this study, we demonstrated that PD-1Ab21 targeted IL-21 to activated T cells and more efficiently induced differentiation of T$_{SCM}$ from antigen-activated CD8$^+$ T cells than recombinant IL-21 in vitro. Flow cytometric analysis and imaging showed that PD-1Ab21 rapidly bound to tumor-specific T cells in peripheral blood and TILs in tumor-bearing mice after systemic administration. The PD-1Ab21 treatment increased the frequency of CD122$^+$Sca1$^+$ cells in CD62L$^{high}$CD44$^{low}$ CD8$^+$ T cell populations, known as T$_{SCM}$, in

spleens and lymph nodes of tumor-bearing mice. This is similar to the intratumoral injection of IL-21, which led to enlarged DLNs with increased naïve lymphocyte numbers and proliferation of activated lymphocytes[46]. In antitumor responses, robust CD8$^+$ T cell priming occurs primarily in DLNs. Once activated, T cells downregulate CD62L or CCR7 expression and egress from DLNs. When the frequency of tumor-specific CD8$^+$ T cells in DLNs of vaccinated tumor-bearing mice decreased to the same level as that of non-vaccinated mice, PD-1Ab21-treated mice still maintained a high frequency of tumor-specific T cells in DLNs, which mostly had the CD62L$^{high}$ phenotype. Furthermore, we identified that PD-1Ab21 treatment biased the differentiation of tumor-specific CD8$^+$ T cells into CD127$^{high}$KLRG-1$^-$ MPECs in tumors relative to anti-PD-1 antibody therapy. MPECs are the effector CD8$^+$ T cells that serve as precursors to a stable pool of memory T cells[38,47]. These data suggest that PD-1Ab21 treatment can generate a pool of tumor-specific memory T cells in secondary lymphoid organs; this pool will self-renew and persistently differentiate into CD62L$^{low/-}$ effector T cells, which egress from DLNs and further develop into MPECs and T$_E$ in tumors. In contrast, anti-PD-1 antibody therapy, or a combination of PD-1 blockade and IL-21, does not have this effect.

T$_{SCM}$ was previously proposed to differentiate directly from naïve cells without passing through an effector phase. CD8$^+$ T$_{SCM}$ cells can be generated in vitro from naïve T cells through the induction of Wnt signaling during T cell priming using either Wnt3A or inhibitors of glycogen synthase kinase-3β (GSK-3β[29]. Further studies show that a combination of IL-7, IL-15, and CD3/CD28 stimulation leads to the generation of CD8$^+$ T$_{SCM}$ from naïve T cells[13]. IL-21 profoundly inhibits T cell differentiation, allowing for the generation of T$_{SCM}$-like T cells[8,11,12,48]. The above methods generate T$_{SCM}$ from naïve T cells. Notch signaling is able to convert activated T cells into T$_{SCM}$-like cells; however, this was achieved by coculturing activated T cells with Notch ligand-expressing stromal cells in the presence of IL-7 and IL-15 or anti-IFN-γ neutralizing antibody for a sustained period of time (11 days)[45]. In contrast, PD-1Ab21 alone can rapidly differentiate activated CD8$^+$ T cells to T$_{SCM}$ in vitro. Moreover, the differentiated T$_{SCM}$ (CD44$^{low}$CD62L$^{high}$) underwent more rounds of division than did the non-differentiated activated cells (CD44$^{high}$CD62L$^{high}$) in the culture. This result demonstrates that the generation of T$_{SCM}$ by PD-1Ab21 involves active cell differentiation through the promotion of T cell proliferation rather than inhibition of cell proliferation and differentiation like Wnt signaling. Therefore, it can be speculated that PD-1Ab21 treatment would be more effective in combination with strategies for T cell priming. This hypothesis is supported by the therapeutic effects of combination therapies with tumor antigen vaccination or anti-Her2/neu antibody.

In summary, we have developed a strategy to improve the therapeutic effect of immune checkpoint blockade by simultaneously targeting cytokines to tumor-reactive T cells to promote the generation of memory T cells in vivo. The results highlight the need to translate this strategy to human cancers. PD-1Ab21 is a single-chain anti-PD-1 antibody, which has a short half-life in vivo. Therefore, a complete human anti-PD-1 antibody fused with IL-21 will be more effective in cancer therapy.

## Methods

**Mice.** Balb/c and C57BL/6 female mice (6–8 week) were purchased from Charles River Experimental Animal (Beijing, China). OT-1 mice having a transgenic TCR specific for H-2Kb and $OVA_{257-264}$ were provided by Dr. Hongyang Wang (Eastern Hepatobiliary Surgery Hospital, Shanghai, China). OT-1/Thy1.1 mice were obtained by backcrossing Thy1.1+ C57BL/6 and OT-1 mice. PD-1-knockout mice were provided by Dr. Lieping Chen (Yale University School of Medicine, New Haven, CT). All mice were maintained in a specific pathogen-free barrier facility at the Institute of Biophysics, CAS. The studies were approved by the Institutional Animal Care and Use Committee of the Institute of Biophysics, CAS and all experiments were conformed to the relevant regulatory standards.

**Cell lines and reagents.** TUBO, cloned from a spontaneous mammary tumor in a Balb/c *neu*-transgenic mouse, was provided by Yang-Xin Fu (University of Texas Southwestern Medical Center, Dallas, TX). Murine colon carcinoma cell line CT26, MC38 and murine metastatic melanoma cell line B16-OVA were provided by Dr. Lieping Chen. TUBO and CT26 were cultured in DMEM medium supplemented with 10% heat-inactivated FBS (PAN), 1% nonessential amino acids (Corning), 1% L-glutamine (Corning), Hepes (10 mM), 1% sodium pyruvate (Corning) and penicillin/streptomycin (10,000 U/ml; Gibco). B16-OVA was cultured in complete RPMI-1640 medium (Hyclone). The pre-B cell line Baf3 (bought from BNCC, China) was cultured in complete RPMI-1640 medium. Anti-PD-1 (G4) and anti-Her2/neu (7.16.4) antibodies were purified from ascites of DKO mice, which were previously treated with intraperitoneal injection of 500 μl liquid paraffin and then intraperitoneally (i.p.) injected a week later with $5 \times 10^6$ G4 hybridoma cells provided by Dr. Lieping Chen or 7.16.4 hybridoma cells provided by Yang-Xin Fu.

**Vector construction and recombinant protein preparation.** The heavy and light chain variable region of anti-PD-1 antibody was amplified by PCR from cDNA of G4 hybridoma, and then cloned into pTT3 vector gifted by Dr. Yingfang Liu (Institute of Biophysics of CAS, Beijing, China) alone or with IL-21 to prepare PD-1Ab or PD-1Ab21. Overlap PCR was used to link the adjacent DNA fragments, and a 3×Flag tag was added to the C-terminus. The IL-21R cDNA was cloned from murine splenocytes and then cloned into pMIgV vector (provided by Lieping Chen) to prepare IL-21RIgFc. Constructs for IL-21Flag and IL-21IgFc were described previously[49]. The primers used were listed in Supplementary Table 1. Proteins were expressed by transient transfection of 293E cell line with the corresponding expression plasmids and purified on anti-Flag M2 affinity gel (Sigma–Aldrich) or protein G column (GE). For antibodies and fusion proteins, the endotoxin level was determined to be lower than 0.2 EU/μg Ab or protein. IL-21-related proteins were quantified with a mouse IL-21 ELISA kit (eBioscience, 88-8210-88). The other proteins or antibodies were measured by Quick Start™ Bradford 1x Dye Reagent (Bio-Rad, #500-0205).

**Size-exclusion chromatography.** A sephadex 200 chromatography column (Sigma–Aldrich) was used to analyze the gel filtration profiles of the purified PD-1Ab21 fusion protein (0.5 μg/ μl) in PBS pH7.4, at a flow rate of 0.5 ml per min. Protein elution was monitored by measuring absorbance at 280 nm.

**Bioassays of PD-1Ab21 fusion protein.** IL-21 bioactivity was tested by measuring proliferation of Baf3, which is an IL-21-dependent pro-B cell line[50]. Baf3 cells were cultured in complete RPMI-1640 medium (Hyclone) supplemented with 2% fetal bovine serum at 37 °C in 5% $CO_2$. Baf3 cells were seeded into a 96-well-plate at $1 \times 10^4$/well, then cultured with murine IL-21 or PD-1Ab21 at the indicated concentrations. Baf3 cells were counted after three days using a cell counter (CountStar).

**Flow cytometry.** The following antibodies were used for the flow cytometry analysis: phycoerythrin (PE)- and Cy7-conjugated anti-CD8 (clone 53-6.7; eBioscience), allophycocyanin (APC)-conjugated anti-CD3 145-2C11; BioLegend), fluorescein isothiocyanate (FITC)-conjugated-CD25 (clone PC 61; eBioscience), APC-anti-CD44 (clone IM7; eBioscience), PE-anti-CD62L (clone M1L-14; Tonbo), PE-anti-CD122 (clone TM-b1; eBioscience), PE-anti-CD127 (clone A7R34; BioLegend), PerCP- and Cy5.5-conjugated-ScaI (clone D7; eBioscience), APC-anti-KLRG1 (clone 2F1; BioLegend), PE-anti-NK1.1 (clone PK136, eBioscience), FITC-anti-PD-1 (clone J43; eBioscience), PE-anti-B7H1 (clone 10F.9G2), PerCP-Cy5.5-anti-CD45 (clone 30-F11; BioLegend), FITC-anti-CD90.1 (clone HIS51;

eBioscience), APC-anti-Flag (clone L5; BioLegend), PE-Cy7-anti-Bcl-2 (clone 10C4; eBioscience), PE-anti-TCF1 (clone 14456S; Cell Signaling Technology), FITC-anti-CD95 (clone SA367H8, BioLegend), PE-anti-IFN-γ (clone XMG1.2; eBioscience), and APC-anti-IL-2 (clone JES6-5H4; eBioscience). For the CFSE dilution assay, T cells were labeled with 5 μM CFSE before being cultured. Dead cells were discriminated with the Fixable Viability Dye eFluor™ 780 (Thermo Fisher Scientific). The stained cells were analyzed with a cytoflex instrument (Beckman coulter). The data analysis was performed using FlowJo X software (Tree Star).

**In vitro binding and blockade.** EG7 cell lines, which were derived from the murine T cell lymphoma EL-4 transfected with cDNA for OVA and constitutively express PD-1, were used to measure the binding and blocking functions of PD-1Ab21 fusion protein. For binding assays, $2 \times 10^5$ EG7 cells were incubated with PD-1Ab21 (0.2 μg in 100 μl) for 30 min at 4 °C with or without pre-incubation of anti-PD-1 (1 μg) for 30 min. The binding of PD-1Ab21 to PD-1 was examined by APC labeled anti-Flag. For blocking assays, $2 \times 10^5$ EG7 cells were incubated with PD-L1IgFc (0.2 μg in 100 μl) for 30 min at 4 °C with or without pre-incubation of PD-1Ab21 (1 μg) for 30 min. The binding of PD-L1IgFc to PD-1 was examined using PE-labeled anti-hIgFc. OT-1 cells activated with $OVA_{257-264}$ peptide in vitro for 2 d were used to detect the binding of PD-1Ab21 to PD-1+ T cells. The binding and blocking assays were performed as described above.

**In vitro differentiation of T cells.** Lymph node cells prepared from OT-1 transgenic mice were stimulated with $OVA_{257-264}$ peptide (0.02 ng/ml) in complete RPMI-1640 medium (Hyclone) complemented with 10% heat-inactivated FBS (PAN), 1% nonessential amino acids (Corning), 1% L-glutamine (Corning), Hepes (10 mM), 1% sodium pyruvate (Corning) and penicillin/streptomycin (10,000 U/ml; Gibco), and 50 μM β-mercaptoethanol (Sigma–Aldrich). After 36–48 h, OT-1 cells were washed and resuspended in complete medium at a concentration of $1 \times 10^6$/ml. IL-2 (Peprotech, 10 ng/ml), equimolar mounts of IL-21 (Peprotech), IL-21IgFc, PD-1Ab or PD-1Ab21were added. After differentiation for 2–3 d, the phenotypes of T cells were analyzed using flow cytometry. Naïve T cells isolated from the lymph node of wild-type (WT) or PD-1-knockout mice were stimulated with anti-CD3 (0.1 μg/ml, Biolegend) and anti-CD28 (0.05 μg/ml, Biolegend) for 40 h, and then differentiated with IL-21 or PD-1Ab21. For the IL-21R blocking assay, IL-21RIgFc was incubated with IL-21 or PD-1Ab21 for 2 h before added to the T cells.

For IL-2/IL-15-induced T cell differentiation, lymph node cells prepared from OT-1 transgenic mice were stimulated with $OVA_{257-264}$ peptide (0.1 ng/ml) for 2 d. Cells were then washed and cultured with IL-2 (10 ng/ml) at a concentration of $3 \times 10^5$ cells/ml or IL-15 (20 ng/ml) at a concentration of $1 \times 10^6$ cells/ml for 4 d.

**T cell sorting.** Single-cell suspensions were blocked with anti-FcR (clone 2.4G2) for 10 min and then stained with antibodies against CD8, CD44, and CD62L for 30 min at 4 °C followed by washing with PBS. Cells were sorted by FACS Aria II Cell Sorter (BD).

**Experiments of tumor therapy.** $2 \times 10^6$ CT26, $5 \times 10^5$ MC38, $1 \times 10^6$ TUBO, or $1 \times 10^6$ B16-OVA tumor cells were subcutaneously (s.c.) injected into the right flank of Balb/c or C57BL/6 mice. In the CT26 tumor model, mice were treated with i.p. injections of 200 μg anti-PD-1 on days 4, 7, 10, or 150 μg PD-1Ab21, or equimolar both PD-1Ab and IL-21 on days 4, 6, 8, and 10 after tumor inoculation. In the MC38 tumor model, mice were treated with i.p. injections of 200 μg anti-PD-1 on days 10, 13, 16, or 150 μg PD-1Ab21, or equimolar both PD-1Ab and IL-21 on days 10, 12, 14, and 16 after tumor inoculation. In the TUBO tumor model, mice were treated with i.p. injections of 200 μg anti-Her2/neu alone on days 16, or in combined with 150 μg PD-1Ab21, or in combined with equimolar both PD-1Ab and IL-21 on days 19, 21, 23, and 25. In the B16-OVA tumor model, $1 \times 10^6$ naïve CD90.1+ OT-1 cells were intravenously (i.v.) transferred into C57BL/6 mice one day before tumor inoculation. Vaccination with poly I:C (50 μg, Sigma) and $OVA_{257-264}$ peptide (50 μg) in a total volume of 200 μl PBS was s.c. administered on day 6. Treatments were administered with i.p. injections of 200 μg anti-PD-1 on days 9, 12, 15, or 150 μg PD-1Ab21, or equimolar both PD-1Ab and IL-21 on days 9, 11, 13, 15, and 17. Tumor length (a) and width (b) were measured and tumor volume was calculated as $(ab^2/2)$.

**PD-1 receptor occupancy assay.** To analyze PD-1Ab21 targeting in vivo, B16-OVA tumor-bearing mice with transferred OT-1 cells were immunized with OVA peptide on day 6, followed by i.p. injection of 200 μg of anti-PD-1 or 100 μg of PD-1Ab21 on day 9 after tumor inoculation. Blood and tumor tissue were collected at the indicated time points after injections of antibody and PD-1AB21. Peripheral blood mononuclear cells and single-cell suspension of tumors were stained with a fluorescent anti-PD-1 antibody. The binding of this antibody to OT-1 cells was analyzed using flow cytometry.

**Preparation of leukocytes from murine tissues.** Spleen and lymph node were ground with a glass slide, and then passed through a 70 μm cell strainer. The spleen cells were subjected to RBC lysis with ACK buffer before antibody staining. FACS buffer (PBS containing 2% FBS) was used to suspend the cells. Tumor tissues were

excised and digested with 1 mg/ml Collagenase IV (Sigma) and 100 mg/ml DNaseI (Sigma) on a shaking table at 37 °C. After 15–20 min, the cell suspensions were passed through a 70-μm cell strainer. The obtained single-cell suspensions were washed once with PBS and suspended with FACS buffer for flow cytometry analysis.

**RNA-seq**. RNA was prepared from $2 \times 10^5$ cells per sample by RNeasy mini kit (Qiagen, 74104). Library construction and sequencing were performed on a BGISEQ-500. SOAPnuke was used to filter low-quality reads (more than 20% of the base qualities are lower than 10), reads with adaptors and reads with unknown bases (N bases more than 5%). Clean reads were mapped to reference using Bowtie2. For the gene expression analysis, the matched reads were calculated and then normalized to RPKM using RESM software. Hierarchical clustering analysis was performed with Morpheus (https://software.broadinstitute.org/morpheus). Principal component analysis (PCA) was performed using the built-in R function prcomp(). Differential expression analysis was performed using Bioconductor package DESeq2.

**ELISA**. The sorted cells were immediately stimulated with PMA (20 ng/ml) and ionomycin (1 μg/ml) for 4 h. IL-2 and IFN-γ in the supernatants were measured by ELISA according to the manufacturer's protocol. The following kits were used: Mouse IL-2 ELISA kit (eBioscience), Mouse IFN-γ ELISA Kit (eBioscience).

**Immunofluorescence staining**. Tumor tissues were embedded with O.C.T. Tissue sections (5 μm) were incubated with primary antibodies overnight at 4 °C after treatment with 0.2% Triton X-100 for 15 min at 4 °C. Then the sections were washed with PBS three times and incubated with secondary antibodies for 2 h at room temperature. Images were captured using a STORM confocal microscope. The following antibodies were used: anti-CD8α (Santa Cruz Biotechnology), anti-Flag (Sigma).

**Statistical analysis**. All data were analyzed using Prism 7.0 Software (GraphPad) and presented as mean ± SEM or ±SD. Statistically significant differences were analyzed by unpaired two-tailed *t*-test and one-way or two-way ANOVA with Tukey's multiple-comparisons test. The details of performed statistical tests and *P*-values are provided in the figure legends.

**Reporting summary**. Further information on research design is available in the Nature Research Reporting Summary linked to this article.

## Data availability

The RNA-seq data have been deposited in the NCBI GEO database under the accession code GSE164093. All the other data supporting the findings of this study are available within the article and its supplementary information files and from the corresponding author upon reasonable request. A reporting summary for this article is available as a supplementary information file. Source data are provided with this paper.

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

## Acknowledgements

We would like to thank Yingfang Liu for help with size-exclusion chromatography. This work was supported by the National Key R&D Program of China (2016YFCCB1303404 and 2019YFC1316100 to S.W.), the Natural Science Foundation of China (91642101 and 81830003 to S.W., 31300728 to Y.L.), the Youth Innovation Promotion Association of the Chinese Academy of Sciences (2015074 to Y.L.), and the Beijing Science and Technology Project under grants (Z181100003818023 to H.H.).

## Author contributions

Y.L. and S.W. designed the experiments, analyzed data, and wrote the paper. Y.L. and Y.C. performed the experiments, analyzed data and wrote the paper. Y.C., Q.H., H.Z., H.L. Y.Z. and H.H. prepared the proteins and Abs. Y.L., M.J. and S.W. analyzed RNA-seq data. Y.L., Y.C., Q.H., M.Y., J.Y. and J.L. performed in vitro experiments. Y.L. and Y.C. performed in vivo experiments. Y.C. performed the immunofluorescence staining and analysis. S.W. and L.C. supervised the research. L.C. provided PD-1$^{-/-}$ mice.

## Competing interests

Pinze Bio-Technique Co. Ltd partly funded this study. S.W., Y.L. and Y.C. are named as co-inventors on pending and issued patents co-owned by Pinze Bio-Technique Co. Ltd and Insititute of Biophysics, CAS. The other authors declare that they have no competing interests.
