## [Peer Review File · Nature Communications]

Reviewers' comments:

Reviewer #1 (Remarks to the Author): expert in T cells and cancer immunotherapy

Yin and colleagues report results on the use of an IL21- antiPD1 fusion molecule in the context of immunotherapy. The authors show that IL21-antiPD1 fusion protein promotes the expansion of memory stem T cells with enhanced cell proliferation. In addition, they show in vivo that such combination is superior to vaccination alone or to concurrent administration of IL21 and antiPD1 in OT1 tumor models and in the context of adoptive T cell transfer. This study also shows that fusion molecule promotes the formation of a memory-like T cell pool endowed with greater early proliferative potential and improved persistence in vivo. Nonetheless, some issues should be addressed:

- The TSCM pool is a T cell subset defined by a specific set of phenotypic markers and TFs; using the expression of CD62L only as a major determinant of such pool is misleading and as such speaking of TSCM in Fig.2 and fig.5 ends up to be incorrect. Please revise the nomenclature or clarify the phenotype in such experimental settings.
- It might be noteworthy to compare the efficiency of induction of TSCM with the currently published protocols (Gattinoni and Cieri). It seems essential to add as a control the "PD-1 alone" treatment in the first figures (especially fig.3 and fig.4).
- Figure 2a: Is the increased percentage of CD44^{lo}/CD62L^{high} observed upon exposure to IL21-antiPD1 fusion molecule statistically significant?
- Is the overall expansion (ie: Abs numbers of cells) of CD44^{lo}/CD62L^{high} cells increased upon exposure to IL21-antiPD1 fusion molecule? This is particularly relevant, considering that IL2 exposure actually induces more proliferation of CD44^{lo}/CD62L^{high} cells than IL21- antiPD1 (Figure 2b).
- Figure 2c shows an increased percentage of CD44^{lo}/CD62L^{high} cells, in sorted CD44^{high}/CD62L^{high} cells, activated in the presence of the IL21- antiPD1 molecule. Again no data are available on the overall expansion of the cells. Furthermore, it would be useful to verify whether the TSCM cells differentiated with the fusion molecule originate only from TCM (CD44^{high}/CD62L^{high}) or also from other cell subsets (CD44^{high}/CD62L^{low}), and most importantly from the population of TN/TSCM itself (CD44^{low}/CD62L^{high}).
- Figure 3 shows results of RNA-seq performed in sorted TN, activated CD44^{high}CD62L^{high}, PD-1Ab21-generated CD44^{low}CD62L^{high} and IL-15 or IL-2-differentiated cells. Data are convincing, although it would be interesting to see results of cells exposed to Anti-PD1 alone or IL-21 alone.

- Lines 150-152: despite the phenotype here described is also present in Naive T cells, cells differentiated from a subset of antigen experienced T cells, cannot be naïve. Please, correct.

- Figure 6 shows 3 in vivo models used to verify the anti-tumor effect of the fusion compounds. The schedule of treatment (already present in Mat Meth) should be added in the figure. A control group inclusive of IL21 alone would be useful. Furthermore, no statistical analysis is shown for the first 2 models.

Minor

- Fig. 1d, the dotted line is not visible in the upper right graph

- Lines 296-298 are possibly too speculative. Self-renewal is not tested in this paper. Please revise

- Figure 4 and 5 could be combined in 1 single figure.

Reviewer #2 (Remarks to the Author): expert in T cell and Ab engineering

Li et al describe an anti-PD1/IL-21 fusion protein and explore in vitro and in vivo effects in comparison with PD1/IL-21 combination. This is a very detailed work with considerable experimental data. Given what is known about IL-21, the rationale behind targeting IL-21 is sound. This work is interesting and timely. The paper is clearly written.

MAJOR COMMENTS

1. The rationale for generating this fusion protein should be explained in the introduction a bit more. IL-21 has been given in the clinic with some sense of MTD. It is moderately, but not particularly toxic. Consequently, it may be possible to give combination of IL-21 and anti-PD1 blockade and just how

much a theoretical advantages of this fusion protein would be should be brought out. Since it is anticipated that the PD1 blocking component should be active too, the authors should be able to establish the kind of systemic concentration needed for anti-PD-1 activity and what bearing this might have (based on IL-21 MTD) on cytokine toxicity. In otherwords, if you have to give so much of the fusion protein to block PD-1 that you get IL-21 toxicity, is it not easier to give both separately so you can titer the relative amount to control toxicity?

2. Figure 1 - demonstration of biological function of IL21, demonstration of biological function of PD1 blockade would be nice to see as part of this characterization.

3. Figure 2 onwards - the authors then show that culture of stimulated T-cells with anti-PD1-IL-21 fusions confers a less differentiated phenotype. This effect is similar but more pronounced than that of just IL-21. The histogrammes of figure 2a (right) should be converted into a form where individual data points can be seen. For example, this can be broken up into 2 or three graphs showing proportion of CD44^{low}CD26^{Lhigh} etc with a scatterplot type graph. A statistical analysis of these data should be offered.

4. Further data here attempting to dissect whether this is a reversion of phenotype or preservation of phenotype has been well covered in the IL-21 literature. Unless there is something fundamentally different that the fusion protein is doing to T-cell biology in vitro (rather than a quantitative effect perhaps due to better approximation), I do not believe these data add much to this work.

5. In Figure 3, similarly, there is work performed to characterize Tscm cells consequent to anti-PD1-IL21 fusion. This is certainly of interest, but the correct control is comparison with IL21 or IL21/anti-PD1 mixture, otherwise we are just likely studying an IL-21 effect without any context of what benefit or unexpected differences the anti-PD1-IL21 fusion might add to transcriptome or phenotype of T-cells. Put in other-words, the assertion (line 178) "Thus, the gene profiling analysis provides evidence that PD-1Ab2a induces conversion of activated CD8+ T-cells back to a memory subset...similar to Tscm" would be much more interesting and relevant if this was contrasted with the effect of IL-21 alone.

6. Figure 4 (a) - these data are interesting characterization, but once again the control (at least) of IL-21 alone is missing. The data should be presented as a scatter type graph so all data points should be seen.

7. Figure 4 (b) and (c) - these data show the performance of adoptively transferred T-cells either generated in IL-2 or anti-PD1-IL-21 fusion. My understanding is that this fusion protein's application

is as a therapeutic rather than a reagent for ex-vivo T-cell generation. Hence, unless these data uncovers some interesting biology particular to the fusion protein, I don't believe this adds much. Without the control of IL-21 cultured T-cells, one can argue that all we are seeing here is an IL-21 effect, which is as expected.

8. Figure 5 (a) the is a direct in vitro comparison between IL-21 and the fusion protein with the conclusion that the fusion protein is approximately 3 times more potent. There are no error bars on the graph which should be shown. A three-fold difference is small and may well be within error range. For the comparison in (b) with IL-21-Fc fusion, only single flow plots with no dose-response curve is shown so any conclusion is not sound. I'm not sure that this is so important to explore, but if done, a dose-response with at least three independent replicates should be performed and error bars shown (perhaps this could be amalgamated into a single dose-response with three conditions).

9. Figure 5(c) shows a separation of stem cell induction after anti-CD3/CD28 vs IL-21 and this is repeated in PD-1 knock-out T-cells. There are a few considerations with these data (a) I only see error bars in the fusion protein condition on the graph on the left; (b) why is the IL-21 dose-response blunted in both experiments, particularly with the PD-1 ko T-cells - it almost looks like IL-21 isn't working at all here even at quite high concentrations; (c) Also, given that the responses separate in both experiments is the assertion that there is a real difference sound? e.g. with the PD-1 ko T-cells, I would have expected the shape of the curve to be the same since as far as these T-cells are concerned, this is just cytokine. Please carefully go over your data and re-design the experiment / provide enough replicates etc to show how much of an effect PD1 binding has. One alternative is to use the PD-1 blocking antibody.

10. Assertion line 228 "However, PD-1Ab21 consistently exhibited more potent anti-tumoural effects through repeated experiments." The data shown doesn't support this assertion. In figure 6(a) for example the anti-PD-1 condition shows 1/5 responses while the fusion protein condition shows 2/5; Supplementary figure 4(a) shows the same numbers of responders in fusion protein vs mixed. There may well be a benefit in these models but the effect is likely relatively small and the experiments need to be powered sufficiently to show a believable difference over chance.

11. As regards figure 6(b), we see 5/5 with the fusion vs 3/5 with the mixture. Again, this could easily be chance as this experiment should be appropriately powered to make the asserted conclusion.

12. Figure 6(c) please show spider plots as per (a) and (b) so growth curves for individual mice can be seen. Please check your data for the highest point in the control since the error bar narrows which is unusual (should widen over time). Please check how your error bars were calculated generally for

this experiment - the error bars are tighter than I would expect given the spider plots presented in (a) and (b).

13. Figure 7(a) This figure might be improved by directly staining for the fusion protein

14. Figure 6(c) and (d) are really important but the crucial control of IL-21 is not included (only vaccine, vaccine + anti-PD1, vaccine + fusion). This makes it really difficult to understand the benefit of the fusion protein as opposed to the just giving a PD-1 and IL-21 combination.

15. Serum concentrations of fusion protein should be measured in the animal models and any toxicity described.

GENERAL COMMENTS:

Figure 1e. Please show all replicates as scatter graph.

Reviewer #1: (Remarks to the Author): expert in T cells and cancer immunotherapy

Yin and colleagues report results on the use of an IL21- antiPD1 fusion molecule in the context of immunotherapy. The authors show that IL21-antiPD1 fusion protein promotes the expansion of memory stem T cells with enhanced cell proliferation. In addition, they show in vivo that such combination is superior to vaccination alone or to concurrent administration of IL21 and antiPD1 in OT1 tumor models and in the context of adoptive T cell transfer. This study also shows that fusion molecule promotes the formation of a memory-like T cell pool endowed with greater early proliferative potential and improved persistence in vivo. Nonetheless, some issues should be addressed:

- The TSCM pool is a T cell subset defined by a specific set of phenotypic markers and TFs; using the expression of CD62L only as a major determinant of such pool is misleading and as such speaking of TSCM in Fig.2 and fig.5 ends up to be incorrect. Please revise the nomenclature or clarify the phenotype in such experimental settings.

Response:

We thank the reviewer for this thoughtful comment and agree that the T_{SCM} pool is a T cell subset defined by a specific set of phenotypic markers and TFs. In Fig.2, we demonstrate that PD-1Ab21 differentiates the activated CD44^{high}CD62L^{high} CD8⁺ T cells back to CD44^{low}CD62L^{high} phenotype through cell proliferation. We named “the PD-1Ab21-generated CD44^{low}CD62L^{high} CD8⁺ T cells” as “naïve-like T cells” in Fig2. Subsequently, we used RNA-seq and flow cytometry to define this cell population by comparing with naïve, activated and other cytokine-differentiated CD8⁺ T cells. RNA-seq and flow cytometry analyses showed that PD-1Ab21-generated CD44^{low}CD62L^{high} T cells expressed high-levels of Ly6a (Sca1-1), Fas (CD95), Pou6f1, Bcl-6, STAT3, and Tcf7(TCF1), which are known to be highly expressed in T_{SCM} or promote memory CD8⁺ T cell formation. To be more rigorous, we re-named “T_{SCM}” as “T_{SCM}-like cells” in Fig.4, which is Fig.5 in the original manuscript, and in related text in the revised manuscript,.

It might be noteworthy to compare the efficiency of induction of TSCM with the currently published protocols (Gattinoni and Cieri). It seems essential to add as a control the “PD-1 alone” treatment in the first figures (especially fig.3 and fig.4).

Response:

Gattinoni and Cieri studied human T_{SCM}, while we studied mouse T cells. Therefore, we did not compare the efficiency of induction of T_{SCM}. Given that anti-PD-1 antibody had no effect on T cell differentiation in vitro, as many papers have shown, we omitted the anti-PD-1 antibody treatment in some experiments. Following the reviewer’s suggestion, we have added the treatment of anti-PD-1 antibody in the study of cell function in Fig.3, which is a combination of original Fig.3 and Fig.4.

Figure 2a: Is the increased percentage of CD44^{lo}/CD62L^{high} observed upon exposure to IL21- antiPD1 fusion molecule statistically significant?

Is the overall expansion (ie: Abs numbers of cells) of CD44^{lo}/CD62L^{high} cells increased upon exposure to IL21-antiPD1 fusion molecule? This is particularly relevant, considering that IL2 exposure actually induces more proliferation of CD44^{lo}/CD62L^{high} cells than IL21- antiPD1 (Figure 2b).

Figure 2c shows an increased percentage of CD44^{lo}/CD62L^{high} cells, in sorted CD44^{high}/CD62L^{high} cells, activated in the presence of the IL21- antiPD1 molecule. Again no data are available on the overall expansion of the cells. Furthermore, it would be useful to verify whether the TSCM cells differentiated with the fusion molecule originate only from TCM (CD44^{high}/CD62L^{high}) or also from other cell subsets (CD44^{high}/CD62L^{low}), and most importantly from the population of TN/TSCM itself (CD44^{low}/CD62L^{high}).

Response:

We thank the reviewer for the thoughtful suggestions. To adequately address these concerns, we (i) repeated the experiment and counted the cells for Fig.2, (ii) replaced the right graph of Fig.2a with a statistical chart to illustrate the statistical differences in the expansion of different cell populations, (iii) replaced the original Fig.2b with a

new figure on the number of CD44^{low}CD62L^{high} cells in different treatment groups, and (iv) added supplementary Fig.1e to show the differentiation of different activated T cell populations.

As shown in Fig.2a (see Figure A below) of the revised manuscript, the frequencies of individual cell populations in the cultures of activated CD8+ T cells with different cytokines or proteins are shown in the histogram. The percentage of CD44^{low}CD62L^{high} cells in culture with PD-1Ab21 was much higher than that in cultures with control, IL-2 or PD-1Ab, and even significantly higher than that in culture with IL-21 or combination of IL21 and PD-1Ab21. As shown in Figure B below, the number of CD44^{low}CD62L^{high} cells in culture with PD-1Ab21 was more than 10 times of that in culture with medium alone. Hence, the overall expansion of CD44^{low}CD62L^{high} was increased upon exposure to PD-1AbIL21.

In the Fig.2b (see the figure below) of the revised manuscript, the activated CD44^{high}CD62L^{high} cells were sorted and cultured in medium alone or in the presence of PD-1Ab21. The cells cultured in medium alone maintained CD44^{high}CD62L^{high} phenotype in 3 days of culture. However, the number of CD44^{low}CD62L^{high} cells gradually increased in the culture with PD-1Ab21 and reached its peak on the second day of cultivation.

Per the reviewer's suggestion, we sorted CD44^{low}CD62L^{high} and CD44^{high}CD62L^{low} population of activated CD8⁺ T cells, and cultured them with PD-1Ab21 or in medium alone for 3 days. As shown in supplementary Fig.1e (see the figure below) of the revised manuscript, PD-1Ab21 could differentiate CD44^{lo}CD62L^{high} cells from CD44^{high}CD62L^{high} cells, but not from CD44^{high}CD62L^{low} cells.

We described these results in the revised manuscript, which were highlighted.

Figure 3 shows results of RNA-seq performed in sorted TN, activated CD44^{high}CD62L^{high}, PD-1Ab21-generated CD44^{low}CD62L^{high} and IL-15 or IL-2-differentiated cells. Data are convincing, although it would be interesting to see results of cells exposed to Anti-PD1 alone or IL-21 alone.

Response:

We thank the reviewer for the supportive comment. The novelty of our fusion protein is to target IL-21 to tumor-specific T cells in vivo. There should be similar effects on differentiation of activated CD8⁺ T cells with IL-21. Additionally, anti-PD-1 antibody had no effect on T cell differentiation in vitro, as many papers have shown. Hence, we did not include experimentation on cells treated with anti-PD-1 alone or IL-21 alone in our original RNA-seq analysis. Following the reviewer's suggestions, we added the cells exposed to PD-1Ab alone and IL-21 alone in the RNA-seq, which is shown in Fig.3a&b and supplementary Fig.2d in the revised manuscript. As expected, CD44^{low}CD62L^{high} T cells generated by PD-1Ab21 or IL-21 clustered together, while PD-1Ab-generated CD44^{high}CD62L^{high} T cells were partitioned in the branch of T_{CM} of the dendrogram. The gene profiling analysis demonstrate that PD-1Ab21 induces conversion of activated CD8⁺ T cells back to a memory subset that is distinct from T_{CM} cells, but similar to T_{SCM}.

Lines 150-152: despite the phenotype here described is also present in Naive T cells, cells differentiated from a subset of antigen experienced T cells, cannot be naïve. Please, correct.

Response:

We thank the reviewer for catching this error in the manuscript and have updated accordingly (highlighted in the revised paper).

Figure 6 shows 3 in vivo models used to verify the anti-tumor effect of the fusion compounds. The schedule of treatment (already present in Mat Meth) should be added in the figure. A control group inclusive of IL21 alone would be useful. Furthermore, no statistical analysis is shown for the first 2 models.

Response:

We thank the reviewer for their points here and have added the schedule of treatments in Fig.6, which is Fig.5 in the revised manuscript. Following the reviewer's recommendation, we used the tumor growth curve figures with the corresponding statistics for all tumor treatment models in main figures (see the figures below) and included the corresponding tumor growth curves for individual mice in the supplementary material (supplementary Fig.4) of the revised manuscript.

We did not set up IL-21 as a control group, because the combinational treatment of IL-21 and PD-1 Ab was not as effective as PD-1Ab21 therapy.

Minor

Fig. 1d, the dotted line is not visible in the upper right graph

Response:

We thank the reviewer for kindly pointing this out in Fig.1d and have made the dotted line visible in the upper right graph (see the figure below).

Lines 296-298 are possibly too speculative. Self-renewal is not tested in this paper.

Please revise

Response:

We thank the reviewer for this thoughtful comment and have modified the statement accordingly (highlighted in the revised paper).

Figure 4 and 5 could be combined in 1 single figure.

Response:

We thank the reviewer for the thoughtful suggestion. We have removed Fig.4b and incorporated Fig.4a within Fig.3 to address the combined suggestions of both reviewers.

Reviewer #2 (Remarks to the Author): expert in T cell and Ab engineering

Li et al describe an anti-PD1/IL-21 fusion protein and explore in vitro and in vivo effects in comparison with PD1/IL-21 combination. This is a very detailed work with considerable experimental data. Given what is known about IL-21, the rationale

behind targeting IL-21 is sound. This work is interesting and timely. The paper is clearly written.

MAJOR COMMENTS

1. The rationale for generating this fusion protein should be explained in the introduction a bit more. IL-21 has been given in the clinic with some sense of MTD. It is moderately, but not particularly toxic. Consequently, it may be possible to give combination of IL-21 and anti-PD1 blockade and just how much a theoretical advantages of this fusion protein would be should be brought out. Since it is anticipated that the PD1 blocking component should be active too, the authors should be able to establish the kind of systemic concentration needed for anti-PD-1 activity and what bearing this might have (based on IL-21 MTD) on cytokine toxicity. In otherwords, if you have to give so much of the fusion protein to block PD-1 that you get IL-21 toxicity, is it not easier to give both separately so you can titer the relative amount to control toxicity

Response:

We thank the reviewer for the thoughtful suggestion. Per the reviewer's suggestion, we've elaborated on the rationale for developing this fusion protein in the introduction (highlighted in the revised paper).

IL-21 is indeed less toxic than other cytokines, such as IL-2. MTD of IL-21 was reported to be 30µg/kg given daily by intravenous bolus for 5 days by 9 days of rest. The treatments of IL-21 were associated with low response rates. Therapeutic application of cytokines for cancer therapy requires that the cytokines reach the tumor and accumulate there to achieve effective concentrations of cytokines within tumors. The half-life of IL-21 is about 1 to 4 hours following intravenous administration. A short half-life limits its exposure and therapeutic efficacy. There may be a failure to achieve effective concentrations of IL-21 on tumor-specific T cells due to its dose-limiting toxicity (DLT). In addition, while activating immune cells to potentiate antitumor immune responses, IL-21 may activate counter-regulatory pathways as

exemplified by IL-2 and IFN- γ which can lead to immune suppression and thereby diminish its therapeutic efficacy. IL-21 has been reported to decrease DC differentiation and antigen presentation, and induce apoptosis of DCs (Wan, CK. et al, Immunity. 2013;38:514; Strengell, M. et al, J Leukoc Biol. 2006;79:1279; Brandt, K. J Invest Dermatol. 2003;121:1379; Brandt, K. Blood. 2003;102:4090.). The novelty of the fusion protein is to concentrate IL-21 to PD-1⁺ T cells in vivo, which can greatly increase the effect of IL-21 on tumor-specific T cells while reducing its side effects. Additionally, fusion of IL-21 to full line of anti-PD-1 antibody can significantly increase the half-life of IL-21.

The conventional dosage of anti-PD-1 antibody is 3mg/kg every 3 or 4 weeks in the clinic. The dosage of 3mg/kg anti-PD-1 antibody is equivalent to a dosage of 300 μ g/kg IL-21, which is little higher than MTD of IL-21 (30 μ g x 5days =150 μ g). In fact, the dosage of 1mg/kg anti-PD-1 antibody was already effective in many clinical trials. Because the antitumor effects of the fusion protein were stronger than that of anti-PD-1 antibody, the dosage of the fusion protein should be lower than that of anti-PD-1 antibody.

2. Figure 1 - demonstration of biological function of IL21, demonstration of biological function of PD1 blockade would be nice to see as part of this characterization.

Response:

The PD-1 antibody in this fusion protein has two roles: (i) to target IL-21 to PD-1⁺ cells by binding to PD-1 receptors on T cells (ii) to block the interaction of PD-1 and PD-L1. These two actions are demonstrated in Fig.1. The biological function of PD-1 blockade has been well demonstrated in many in vivo studies. There is no feasible assay to test the biological function of PD-1 blockade in vitro.

3. Figure 2 onwards - the authors then show that culture of stimulated T-cells with anti-PD1-IL-21 fusions confers a less differentiated phenotype. This effect is similar but more pronounced than that of just IL-21. The histogrammes of figure 2a (right)

should be converted into a form where individual data points can be seen. For example, this can be broken up into 2 or three graphs showing proportion of CD44^{low}CD26L^{high} etc with a scatterplot type graph. A statistical analysis of these data should be offered.

Response:

We thank the reviewer for their well-advised suggestions. We replaced the right graph of Fig.2a with a scatterplot type graph that shows the frequency of individual cell populations. The statistical analysis of the frequency of CD44^{low}CD62L^{high} cell population was offered in the graph. (see the figure below)

4. Further data here attempting to dissect whether this is a reversion of phenotype or preservation of phenotype has been well covered in the IL-21 literature. Unless there is something fundamentally different that the fusion protein is doing to T-cell biology in vitro (rather than a quantitative effect perhaps due to better approximation), I do not believe these data add much to this work.

Response:

While multiple papers have demonstrated that IL-21 induce differentiation of human T_{SCM}, few show induction of mouse T_{SCM} differentiation by IL-21. Hinrichs G.S reported that IL-21 suppressed antigen-induced differentiation of naïve CD8⁺ T cells into effector T cells (Blood, 2008). However, there are no research reports on whether IL-21 can reverse activated (CD44^{high}CD62L^{high}) T cells to T_{SCM}-like cells with

CD44^{low}CD62L^{high} phenotype. Therefore, we believe that the results in Fig.2 are of significant value.

5. In Figure 3, similarly, there is work performed to characterize Tscm cells consequent to anti-PD1-IL21 fusion. This is certainly of interest, but the correct control is comparison with IL21 or IL21/anti-PD1 mixture, otherwise we are just likely studying an IL-21 effect without any context of what benefit or unexpected differences the anti-PD1-IL21 fusion might add to transcriptome or phenotype of T-cells. Put in other-words, the assertion (line 178) "Thus, the gene profiling analysis provides evidence that PD-1Ab2a induces conversion of activated CD8+ T-cells back to a memory subset...similar to Tscm" would be much more interesting and relevant if this was contrasted with the effect of IL-21 alone.

Response:

We thank the reviewer for the comment. The novelty of our fusion protein PD-1Ab21 in this study is to target IL-21 to tumor-specific T cells in vivo. PD-1Ab21 has similar effects on differentiation of activated CD8⁺ T cells with IL-21 in vitro. Hence, we did not include cells treated with IL-21 alone or mixture of IL-21 and anti-PD-1 antibody in our original RNA-seq analysis. Following the suggestions of both reviewers, we added the cells exposed to PD-1Ab alone and IL-21 alone in the RNA-seq, which is shown in Fig.3a&b and supplementary Fig.2d in the revised manuscript. As expected, CD44^{low}CD62L^{high} T cells generated by PD-1Ab21 or IL-21 clustered together, while PD-1Ab-generated CD44^{high}CD62L^{high} T cells were partitioned in the branch of T_{CM} of the dendrogram. The gene profiling analysis demonstrate that PD-1Ab21 induces conversion of activated CD8⁺ T cells back to a memory subset that is distinct from T_{CM} cells, but similar to T_{SCM}.

6. Figure 4 (a) - these data are interesting characterization, but once again the control (at least) of IL-21 alone is missing. The data should be presented as a scatter type graph so all data points should be seen.

Response:

We thank the reviewer for their input and have repeated the experiment including the sorted IL-21-differentiated CD44^{low}CD62L^{high} cells. Each group of differentiated cells had three replicates. According to the reviewer's suggestion, the data was presented as a scatter type graph with visible data points. The results are shown in Fig.3d of the revised manuscript (see the figure below).

7. Figure 4 (b) and (c) - these data show the performance of adoptively transferred T-cells either generated in IL-2 or anti-PD1-IL-21 fusion. My understanding is that this fusion protein's application is as a therapeutic rather than a reagent for ex-vivo T-cell generation. Hence, unless these data uncovers some interesting biology particular to the fusion protein, I don't believe this adds much. Without the control of IL-21 cultured T-cells, one can argue that all we are seeing here is an IL-21 effect, which is as expected.

Response:

We agree with the reviewer's comments and have (i) removed Fig.4b and 4c (ii) incorporated Fig.4a within Fig.3 in the revised manuscript.

8. Figure 5 (a) the is a direct in vitro comparison between IL-21 and the fusion protein with the conclusion that the fusion protein is approximately 3 times more potent. There are no error bars on the graph which should be shown. A three-fold difference is small and may well be within error range. For the comparison in (b) with IL-21-Fc fusion, only single flow plots with no dose-response curve is shown so any conclusion is not sound. I'm not sure that this is so important to explore, but if done, a

dose-response with at least three independent replicates should be performed and error bars shown (perhaps this could be amalgamated into a single dose-response with three conditions).

Response:

We thank the reviewer for the thoughtful comment. In our original experiments, two replicate wells were established for each concentration of IL-21 or PD-1Ab21. Error bars on the graph were too small to be seen. We repeated this experiment with three replicate wells for each concentration and presented the data by using color curves with black error bars in Fig. 4 in the revised manuscript (see the figure below).

Additionally, we agree with the reviewer that the comparison with IL-21-Fc fusion is not so important to explore and have removed this data from the main figure and kept the flow plots with different doses of proteins in the supplementary material.

9. Figure 5(c) shows a separation of stem cell induction after anti-CD3/CD28 vs IL-21 and this is repeated in PD-1 knock-out T-cells. There are a few considerations with these data (a) I only see error bars in the fusion protein condition on the graph on the left; (b) why is the IL-21 dose-response blunted in both experiments, particularly with the PD-1 ko T-cells - it almost looks like IL-21 isn't working at all here even at quite high concentrations; (c) Also, given that the responses separate in both experiments is the assertion that there is a real difference sound? e.g. with the PD-1 ko T-cells, I would have expected the shape of the curve to be the same since as far as these T-cells are concerned, this is just cytokine. Please carefully go over your data and re-design the experiment / provide enough replicates etc to show how much

of an effect PD1 binding has. One alternative is to use the PD-1 blocking antibody.

Response:

We thank the reviewer for their thoughtful comment and have carefully repeated this experiment with three replicate wells for each concentration. Most of the error bars were too small to be seen in the original figures. To illustrate the results more clearly, we presented the data by using color curves with black error bars in Fig. 4 in the revised manuscript. Because the effect of IL-21 on CD44^{low}CD62L^{high} cell differentiation was less than that of PD-1Ab21, the dose-response curve of IL-21 was blunter. But, the frequency of CD44^{low}CD62L^{high} cells were significantly increased with the increase of IL-21 dose. The wild-type and KO T cells were unlikely to be activated very synchronously. Because the activated KO T cells had a little higher proportion of CD44^{low}CD62L^{high} cells than the wild-type T cells at the beginning of cell differentiation, the dose-response curve of IL-21 was blunter in the differentiation of KO T cells.

We have attempted to use anti-PD-1 blocking antibody in this experiment. But, the test was difficult to do. Probably because the affinity difference between the blocking antibody and PD-1Ab21 is not large enough, and their binding is dynamic. In addition, the effects of cytokines are usually fast.

10. Assertion line 228 "However, PD-1Ab21 consistently exhibited more potent anti-tumoural effects through repeated experiments." The data shown doesn't support this assertion. In figure 6(a) for example the anti-PD-1 condition shows 1/5 responses while the fusion protein condition shows 2/5; Supplementary figure 4(a) shows the same numbers of responders in fusion protein vs mixed. There may well be a benefit in these models but the effect is likely relatively small and the experiments need to be powered sufficiently to show a believable difference over chance.

Response:

CT26 tumor model responded variably to PD-1 blockade across the reported studies. In our repeated CT26 tumor treatment experiments, the response of mice to the treatments was always very different in a group. However, PD-1Ab21 treatment

always caused the largest number of tumor regression in each experiment. To further demonstrate the consistency in therapeutic effect, we repeated this therapy using an MC38 tumor, which is another tumor model that responds to PD-1 blockade. In the MC38 tumor model, PD-1Ab21 still has the best therapeutic effect on MC38 tumor. This result is included in included in Fig.5a and Supplementary Fig.4a of the revised manuscript. (see the figures below)

Per the reviewer’s suggestion, we modified the referenced statement to read “PD-1Ab21 exhibited stronger anti-tumor effects through repeated experiments” (highlighted in the revised manuscript).

11. As regards figure 6(b), we see 5/5 with the fusion vs 3/5 with the mixture. Again,

this could easily be chance as this experiment should be appropriately powered to make the asserted conclusion.

Response:

In our previous repeated experiments, we performed two injections of anti-neu antibody. Those treatments had good therapeutic effect on TUBO tumor. So, the differences between anti-neu treatment group and other treatments were not large enough. During the revision of this manuscript, we repeated this treatment experiment with one injection of anti-neu. Although the combinational therapy of anti-neu and PD-1Ab21 could not cause all tumor regression, the therapeutic effect of the combination with PD-1Ab21 was significantly better than that of anti-neu alone or the combination with PD-1Ab and IL-21. This result is shown in Fig.5b and Supplementary Fig.4b of the revised manuscript. (see the figures below)

12. Figure 6(c) please show spider plots as per (a) and (b) so growth curves for individual mice can be seen. Please check your data for the highest point in the control since the error bar narrows which is unusual (should widen over time). Please check how your error bars were calculated generally for this experiment - the error bars are tighter than I would expect given the spider plots presented in (a) and (b).

Response:

For the original Figure 6(c), as shown in figures below, the tumor in one mouse grew faster than the tumors in other mice in the control group. This mouse died at the last

time point. Therefore, the error bars become narrow at the highest point along the growth curve in the control group.

To address the combined suggestions of the reviewers for this figure, we used the tumor growth curve figures with the corresponding statistics for all tumor treatment models in main figures (Fig.5) and included the corresponding tumor growth curves for individual mice in the supplementary material (supplementary Fig.4) in the revised manuscript.

13. Figure 7(a) This figure might be improved by directly staining for the fusion protein

Response:

Given that the assay for PD-1-receptor occupancy has been widely used in studies of anti-PD-1 antibody therapy, we detected the targeting of PD-1Ab21 to tumor-specific T cells in vivo by assessing PD-1-receptor occupancy on OT-1 T cells in single cell suspension of enzymatically digested tumor tissues. Additionally, we also directly stained the fusion protein on tumor-infiltrating CD8⁺ T cells in tumors by double staining with anti-CD8 and anti-flag antibodies in an immunofluorescence analysis (Supplementary Fig.6).

We had also directly stained the fusion protein that bound to OT-1 T cells in peripheral blood and tumors by flow cytometry. After injection, the fusion protein

could be clearly detected on OT-1 T cells, but not on non-OT-1 CD8⁺ T cells in peripheral blood with fluorescence-labeled anti-flag antibody by flow cytometry. But the sensitivity of the detection was too low (see figure below).

Figure. Detection of PD-1Ab21 bound to OT-1 T cells in peripheral blood. B16-OVA tumor-bearing mice with transferred OT-1 cells were immunized with poly I:C and OVA peptide on day 6, followed by i.p. injections of PD-1Ab21 on day 9 after tumor inoculation. Blood was collected 30 minute and 6 hours after protein injection. Lymphocytes were isolated and stained with anti-flag antibody. OT-1 T cells (left) and non-OT-1 CD8⁺ T cells (right) were gated. Lymphocytes isolated from mice without injection of protein were used as control (No protein).

14. Figure 6(c) and (d) are really important but the crucial control of IL-21 is not included (only vaccine, vaccine + anti-PD1, vaccine + fusion). This makes it really difficult to understand the benefit of the fusion protein as opposed to the just giving a (s PD-1 and IL-21 combination.

Reply:

The figures referenced should be the original Fig.7c and 7d because Fig.6d did not exist in the original manuscript. Due to the large number of samples, we did not perform lymphocyte analysis across all treatment groups in one single experiment in the original manuscript. In order to address the reviewer's comment, we repeated the lymphocyte analysis across all treatment groups. The results are shown in Fig.6c & 6d and supplementary Fig.7 in the revised manuscript.

15. Serum concentrations of fusion protein should be measured in the animal models and any toxicity described.

In the beginning of the animal studies, we measured the concentrations of the injected fusion proteins in the serum of injected mice (see figure below). Because the fusion protein was constructed with single chain antibody, the half-life of the fusion protein was short. After intraperitoneal injections, the fusion protein could be detected in the serum at 12 hours, but not at 24 hours.

To detect possible toxic reactions caused by fusion protein treatments, we measured the inflammatory cytokines IL-6, TNF α , IFN- γ , IL-1b in the serum of treated tumor-bearing mice. However, they could not be detected on the second day after completing the treatment.

Figure. *Detection of PD-1Ab21 concentrations in serum.* 150 μ g PD-1Ab21 fusion protein was intraperitoneally injected into tumor-bearing mice. Blood was collected at the indicated time after injection. IL-21 was detected in serum by ELISA.

GENERAL COMMENTS:

Figure 1e. Please show all replicates as scatter graph.

Response:

In the original Fig.1e, we calculated the cell proliferation rate to show the biological activity of PD-1Ab21, but the error bars were large. We repeated the experiments and used the cell numbers to display the cell proliferation more intuitively in the revised manuscript (see figure below). Three independent replicates are displayed in a scatter graph.

REVIEWER COMMENTS

Reviewer #1 (Remarks to the Author):

the authors properly addressed all issues raised by the reviewer.

Reviewer #3 (Replacement for Reviewer#2, Remarks to the Author):

The authors present an extensively revised and improved manuscript.

In regard to their responses to the comments of reviewer 2:

In response to the comment 1, the authors additionally elaborated in the introduction on the rationale of the targeted delivery of IL-21. This supports the clarification of the concept, but does not directly address the main point of the comment, i.e. the fact that due to the low MTD of IL-21 (30 μ g/kg), the dose of antibody-fusion protein systemically applicable would be considerably lower than the antibody dose conventionally used for PD-1 blocking (3mg/kg). This raises the question if a separate application of antibody and cytokine would not be more effective, since individual optimized dosing would be possible. Indeed, in theory, the application of the fusion protein at the MTD of IL-21 would reduce the antibody dose conventionally used by factor 20 (we have here two IL-21 molecules per antibody molecule). The argument in the rebuttal letter that the difference is rather little is not convincingly supported by the calculation and numbers presented. The MTD is not to be estimated as the sum of consecutive daily cytokine dose applications. Daily treatment with the cytokine (short half-life) is not expected to be cumulative.

In the comment 2, the reviewer suggested to complement the characterization of the fusion protein by showing how PD-1 inhibition affects T cell biology. The authors argue that there is no feasible assay to test the biological function of PD-1 blockade in vitro. Since the effect of PD-1 blocking on T cell activity could be easily shown in vitro e.g. by the enhancement in proliferation of activated T cells or IL-2 release in presence of the anti-PD-1 Ab, there must have been some kind of misinterpretation of the comment. However, functionality of the individual components of the fusion protein i.e. cytokine activity of IL-21 and binding/blocking capacity of anti-PD-1 antibody is sufficiently demonstrated in Fig.1. Thus, additional characterization could be considered optional.

The comments 3-7 were addressed satisfactorily.

The comments 8-9 were addressed by the authors as requested by the reviewer. However, here I'd like to add, that the concentration of the proteins in Fig.4 should be indicated in nM and not ng/ml. The MW of the proteins differ and the comparison should be done on an equimolar basis of IL-21 and presented this way. Also, in order to show the variability of the assays, error bars presented as SD and not as SEM would be preferable. Furthermore, it cannot be excluded that the stronger effect observed for the fusion protein might be related to oligomerization of IL-21 (HPLC data Fig.1c). Thus, to address the contribution of targeting to the effect, the use of a control Ab-IL-21 (same recombinant fusion protein format with unrelated Ab specificity) would be a good option.

The comments 10-12 of reviewer 2 in regard to the animal experiments were addressed only partially by the authors. Tumor growth curves for individual mice were provided. It is striking, that the error bars of the group presentation do not always reflect the impression from the graph with the individual mouse tumor growth curves. There seems to be a high variability in the response of individual mice to the fusion protein treatment that is not perceived in the group presentation. Please, refer to the calculation of error bars and statistics for clarification. Were always all mice of the group considered? Also, the authors indicate in the results (line 226-227) that they used "equimolar mixture of PD-1Ab and IL-21, or PD-1Ab21 alone" in the animal experiments. However, in material & methods (page 21) it is indicated, that the same amount of each protein (150µg) was injected. Since the MW of the proteins are not the same, this would not be an equimolar treatment. Please, clarify.

The comments 13-16 were addressed satisfactorily.

Additional comments

- 1) Line 227-229: the statement is only accurate for the MC38 model, but not for the CT26 model.
- 2)The reference for Fig.3B is missing in the text of the manuscript.

Reviewer #1 (Remarks to the Author):

the authors properly addressed all issues raised by the reviewer.

Response:

We thank the reviewers for their insightful comments in the first review which greatly helped us improve this manuscript.

Reviewer #3 (Replacement for Reviewer#2, Remarks to the Author):

The authors present an extensively revised and improved manuscript.

In regard to their responses to the comments of reviewer 2:

In response to the comment 1, the authors additionally elaborated in the introduction on the rational of the targeted delivery of IL-21. This supports the clarification of the concept, but does not directly address the main point of the comment, i.e. the fact that due to the low MTD of IL-21 (30µg/kg), the dose of antibody-fusion protein systemically applicable would be considerably lower than the antibody dose conventionally used for PD-1 blocking (3mg/kg). This raises the question if a separate application of antibody and cytokine would not be more effective, since individual optimized dosing would be possible. Indeed, in theory, the application of the fusion protein at the MTD of IL-21 would reduce the antibody dose conventionally used by factor 20 (we have here two IL-21 molecules per antibody molecule). The argument in the rebuttal letter that the difference is rather little is not convincingly supported by the calculation and numbers presented. The MTD is not to be estimated as the sum of consecutive daily cytokine dose applications. Daily treatment with the cytokine (short half-life) is not expected to be cumulative.

Response:

We thank the reviewer for the thoughtful comments and agree that it is inappropriate to estimate the MTD of IL-21 as the sum of consecutive daily dose applications. In

theory, the application of the fusion protein at the MTD (30 μ g/kg) of IL-21 would reduce the antibody dose conventionally used (3mg/kg) by factor 20. However, the fusion protein should preferentially target to PD-1⁺ T cells, helping to spare other cells. Therefore, the fusion protein should increase the MTD of its IL-21. We recognize that several clinical trials have indeed shown that the immunocytokines composed of IL-2 or IL-12 increases MTD of the cytokines within them. For example, the systemic administration of IL-12 at doses of 0.5 μ g/kg was associated with a number of significant toxicities, while the MTD of immunocytokine huBC1-IL12, composed of two IL-12 heterodimers fused with an anti-ED-B antibody (BC1), was established at 15 μ g/kg (equivalent to 7.5 μ g/kg of IL-12) in a phase I study. The MTD of another similar immunocytokine NHS-IL-12 was 16.8 μ g/kg. IL-2 was conventionally administered at either 7.2 x10⁵ U/kg (high-dose) or 7.2 x10⁴U/kg (low-dose). The high-dose IL-2 is associated with severe life-threatening side effects that require hospitalization for therapy. The MTD of immunocytokine hu14.18-IL2, composed of 14.18 anti-GD2 antibody linked to two molecules IL-2, was established at 12 μ g/m² /daily (1 mg of the fusion protein contains approximately 3x10⁶ U IL-2). The MTD of hu14.18-IL-2 is approximately a high dose of IL-2. However, the dose-limiting toxicities of hu14.18-IL-2 at MTD is similar to that of low-dose IL-2.

IL-2 binds to its trimeric receptor with very high affinity (K_d: ~10pM), which is much higher than most antibodies. Hence, the biodistribution of IL-2-based immunocytokines is likely to be mainly governed by IL-2 moiety rather than by antibodies. As a result, the systemic dose-limiting side effects related to IL-2 are not dramatically ameliorated by attempted antibody targeting. The affinity of IL-21 binding its receptor is lower than that of IL-2 and IL12. Therefore, compared to IL-2 and IL-12, IL-21-based immunocytokines should more significantly increase the therapeutic index of IL-21 payload.

Indeed, their doses can be individually optimized to achieve the best synergistic therapeutic effect when antibodies and cytokines were administered in combination as separate agents. However, the optimization cannot increase the MTD of cytokines. One of the major limitations for cytokine therapy is that the cytokine-caused adverse

effects prevent dose escalation for therapeutically active regimens. The preclinical and clinical studies have shown that the immunocytokines can increase the MTD of cytokines within them by focusing the biological effects of cytokine signaling onto a rather specific cell population.

In this manuscript, we just demonstrated that IL-21 can be targeted to tumor-reactive T cells by fusion IL-21 to anti-PD-1 antibody to improve the therapeutic effects of immune checkpoint blockade. Currently, we are developing human fusion protein composed of a full humanized anti-PD-1 antibody and IL-21. We have taken several measures to improve the pharmacodynamics, including increasing the affinity of anti-PD-1 antibody and constructing the fusion proteins containing one or two IL-21 molecules.

In the comment 2, the reviewer suggested to complement the characterization of the fusion protein by showing how PD-1 inhibition affects T cell biology. The authors argue that there is no feasible assay to test the biological function of PD-1 blockade in vitro. Since the effect of PD-1 blocking on T cell activity could be easily shown in vitro e.g. by the enhancement in proliferation of activated T cells or IL- 2 release in presence of the anti-PD-1 Ab, there must have been some kind of misinterpretation of the comment. However, functionality of the individual components of the fusion protein i.e. cytokine activity of IL-21 and binding/blocking capacity of anti-PD-1 antibody is sufficiently demonstrated in Fig.1. Thus, additional characterization could be considered optional.

Response:

We agree that the effect of PD-1 blocking on T cell activity could be tested in vitro by the enhancement in proliferation of activated T cells or cytokine production. But, in these assays, IL-21 of the fusion protein also has effects on T cell proliferation and cytokine production. It is not easy to distinguish the respective effects of PD-1 inhibition and IL-21.

The comments 3-7 were addressed satisfactorily.

The comments 8-9 were addressed by the authors as requested by the reviewer. However, here I'd like to add, that the concentration of the proteins in Fig.4 should be indicated in nM and not ng/ml. The MW of the proteins differ and the comparison should be done on an equimolar basis of IL-21 and presented this way. Also, in order to show the variability of the assays, error bars presented as SD and not as SEM would be preferable. Furthermore, it cannot be excluded that the stronger effect observed for the fusion protein might be related to oligomerization of IL-21 (HPLC data Fig.1c). Thus, to address the contribution of targeting to the effect, the use of a control Ab-IL-21 (same recombinant fusion protein format with unrelated Ab specificity) would be a good option.

Response:

We thank the reviewer for the thoughtful suggestions. In this assay, we quantified IL-21 and fusion protein by detecting IL-21 with an ELISA kit for IL-21. So, the concentrations of the proteins in Fig.4 were the mass concentrations of IL-21 and not the fusion protein. We apologize for not stating this in our manuscript.

We agree with the reviewer that the stronger effect observed for the fusion protein might be related to oligomerization of IL-21. IL-21 within the fusion protein is a dimer. IL-21IgFc was indeed more effective at stimulating T_{SCM} differentiation than was recombinant monomer IL-21 (Supplementary Fig. 3a). Per the reviewer's suggestion, we repeated the experiments in Fig.4 by using IL-21IgFc as a control. Meanwhile, we compared the dose-dependent effects of PD-1Ab21 and IL-21IgFc with the molality of the proteins. The results were shown in Fig.4 of the revised manuscript. The error bars were presented as SD. The modified statements were highlighted in revised manuscript.

The comments 10-12 of reviewer 2 in regard to the animal experiments were addressed only partially by the authors. Tumor growth curves for individual mice were provided. It is striking, that the error bars of the group presentation do not always reflect the impression from the graph with the individual mouse tumor growth

curves. There seems to be a high variability in the response of individual mice to the fusion protein treatment that is not perceived in the group presentation. Please, refer to the calculation of error bars and statistics for clarification. Were always all mice of the group considered? Also, the authors indicate in the results (line 226-227) that they used “equimolar mixture of PD-1Ab and IL-21, or PD-1Ab21 alone” in the animal experiments. However, in material & methods (page 21) it is indicated, that the same amount of each protein (150µg) was injected. Since the MW of the proteins are not the same, this would not be an equimolar treatment. Please, clarify.

Response:

Tumor growth curves for individual mice were presented in a longer period of time. Some tumors that shrunk early grew back in the fusion protein treatment group. So, there seems to be a high variability in the response of individual mice to the fusion protein treatment. Since mice in other treatment groups began dying at the early time points, the tumor growth curve figures with the corresponding statistics were presented in a shorter period of time. For example, in the experiment of CT26 tumor model, all mice in the fusion protein treatment group survived for more than 22 days, while some mice in other groups were died by that date. At this time point, the shrunk tumor had not grown up in the fusion protein treatment group, while one mouse had not tumor and the other mice had large tumor in anti-PD-1 antibody-treated group (see the figure below). So, the statistical difference of tumor size between mice treated with the fusion protein was smaller than that between mice treated with anti-PD-1 antibody. In the CT26 tumor growth curve with the corresponding statistics in Fig.5A, some mice were not included at the last two time points because some mice had died. The original data of CT26 tumor growth graph in Fig5A is presented in the below table.

We thank the reviewer for catching the errors in the description of experimental methods for tumor therapy. Mice were treated with i.p. injections of 200 µg anti-PD-1 antibody or 150 µg PD-1Ab21, or equimolar both PD-1Ab and IL-21. We have updated this description in the revised manuscript, which was highlighted.

CT26	Ctr					anti-PD-1					PD-1Ab + IL-21					PD-1Ab21				
Days																				
0	0	0	0	0	0	0	0	0	0	0	0	0	0	0	0	0	0	0	0	0
4	41.5	22.7	18.9	34.4	22.3	25.3	32	28.2	29.6	35.2	25.3	29.7	29.7	27.2	34.4	33.6	28.7	36	32	32
7	150	117	97.2	125.3	94.1	125	109.7	85.7	90.4	100.7	124.9	97.5	71.5	124.2	129	144.2	118.8	138.9	135.2	68.5
10	401.6	414.7	303.4	378.5	337.6	137.3	393.3	204.8	258.6	171.4	410.8	314	262.8	326.1	325.1	433.4	277.2	329.1	325.1	173.2
13	820	930.3	670.7	861.2	771.5	119.2	758.9	526.9	474.6	482.8	966.3	657.3	580	672.5	747	740.7	231.9	424.1	410.8	128.3
16	1248.4	1482.6	1014.8	1373.6	1331.7	58.8	1137.1	930.9	659	667	1511		834.2	1173.2	1166.9	817.2	186.3	424.1	364	129.5
19	1709.5	2353.4	1664.7	2178.1	1932.9	0	1901.3	1640.2	960.7	1146.9	2375.1		1385.1	1940.7	1896.1	1163.3	171.1	595.5	326.1	77.1
22	2258.4		2084.5		2662.6	0	2713.3	2051.4	1294.7	1617.4			1771.9	2800.5	2663.7	1295.7	51.9	740.8	336.2	37

The comments 13-16 were addressed satisfactorily.

Additional comments

1) Line 227-229: the statement is only accurate for the MC38 model, but not for the CT26 model.

Response:

We thank the reviewer for catching this inaccurate description in the manuscript and have updated accordingly (highlighted in the revised paper).

2)The reference for Fig.3B is missing in the text of the manuscript.

Response:

We thank the reviewer for kindly pointing out this mistake and have added the reference for Fig.3B in the revised manuscript.

REVIEWERS' COMMENTS

Reviewer #3 (Remarks to the Author):

In regard to the comment 1, the authors wrote: "The preclinical and clinical studies have shown that the immunocytokines can increase the MTD of cytokines within them by focusing the biological effects of cytokine signalling onto a rather specific cell population".

There are no references from the literature indicated to support this statement. There is also no rational explanation from the authors to support this theory. Why is the cytokine to be expected to be less systemically active when it is part of the fusion protein? Why should the maximum tolerated dose be enhanced by targeting, i.e. the systemic toxicity be reduced, especially if a longer half-life of the cytokine due to the fusion to the antibody might be expected?

It seems to me that targeting the cytokine in form of an antibody-fusion protein to a particular cell population in the tumor could be expected to lead to enhanced local concentrations and/or mechanisms of action, thus the dose applied systemically might be reduced and consequently the risk of adverse events.

It is beyond the scope of the manuscript, but the authors might want to address this fundamental question in future in vivo studies.

The other comments were properly addressed.

Further amendments of the manuscript are not required.